# Accelerating the prediction and discovery of peptide hydrogels with human-in-the-loop

Tengyan Xu[1,2,6], Jiaqi Wang[3,4,5,6], Shuang Zhao[5,6], Dinghao Chen[1,6], Hongyue Zhang[1], Yu Fang [1], Nan Kong[1], Ziao Zhou[1], Wenbin Li [3,4,5] ✉ & Huaimin Wang [1,2,3] ✉

The amino acid sequences of peptides determine their self-assembling properties. Accurate prediction of peptidic hydrogel formation, however, remains a challenging task. This work describes an interactive approach involving the mutual information exchange between experiment and machine learning for robust prediction and design of (tetra)peptide hydrogels. We chemically synthesize more than 160 natural tetrapeptides and evaluate their hydrogel-forming ability, and then employ machine learning-experiment iterative loops to improve the accuracy of the gelation prediction. We construct a score function coupling the aggregation propensity, hydrophobicity, and gelation corrector $C_g$, and generate an 8,000-sequence library, within which the success rate of predicting hydrogel formation reaches 87.1%. Notably, the de novo-designed peptide hydrogel selected from this work boosts the immune response of the receptor binding domain of SARS-CoV-2 in the mice model. Our approach taps into the potential of machine learning for predicting peptide hydrogelator and significantly expands the scope of natural peptide hydrogels.

Hydrogel, an important class of soft materials, is formed from a self-assembled matrix that immobilizes water. Hydrogels have attracted increasing attention in various research fields because they mimic properties in natural systems such as the bodies of jellyfish, the cornea in the eye, and even the condensed chromatins in the cell nucleus[1,2]. Inspired by natural self-assembled functional materials (high-order assemblies of proteins), considerable attention has been focused on hydrogels formed by peptides because of their high biocompatibility[3–7], low immunogenicity[8–10], and similarity to the extracellular matrix[11–14]. To date, peptidic hydrogels have been widely used in materials science[15–18], biomedicine[19–22], and semiconductors[23–25]. However, the current design capability fails to meet the growing demand for neoteric peptidic hydrogels since the

existing inefficient methods still rely on amino acid sequences that derive from natural proteins, professional experience in the peptide field, or laboratory discoveries by serendipity[26–28]. Therefore, accurate prediction of hydrogel formation and de novo design of peptidic hydrogels emerge as of great significance to broaden the available hydrogel-forming peptide library.

To better understand the self-assembly behaviors of peptides in forming hydrogels and the resulting morphologies, coarse-grained molecular dynamics (CGMD) has been employed to model peptide self-assembly[29–32]. Ulijn and Tuttle's groups recently developed a useful approach to provide valuable design rules for overcoming the limitation of serendipity in discovering aggregation or self-assembly in dipeptide and tripeptide systems[33,34]. However, molecular dynamics

[1]Department of Chemistry, School of Science, Westlake University, 18 Shilongshan Road, Hangzhou 310024 Zhejiang Province, China. [2]Institute of Natural Sciences, Westlake Institute for Advanced Study, 18 Shilongshan Road, Hangzhou 310024 Zhejiang Province, China. [3]Research Center for the Industries of the Future, Westlake University, No. 600 Dunyu Road, Sandun Town, Xihu District, Hangzhou 310030 Zhejiang Province, China. [4]Institute of Advanced Technology, Westlake Institute for Advanced Study, 18 Shilongshan Road, Hangzhou 310024 Zhejiang Province, China. [5]School of Engineering, Westlake University, 18 Shilongshan Road, Hangzhou 310024 Zhejiang Province, China. [6]These authors contributed equally: Tengyan Xu, Jiaqi Wang, Shuang Zhao, Dinghao Chen. ✉e-mail: liwenbin@westlake.edu.cn; wanghuaimin@westlake.edu.cn

(MD) simulations of selected peptides could only give the information (*e.g.*, aggregation propensity, acronymized as AP) for predicting new peptides that derive from the original ones. Importantly, due to the enormous sequence quantities of peptides, brute-force MD is becoming increasingly intractable for investigating the hydrogel formation ability of longer-chain peptides[33,35,36]. To the best of our knowledge, systematic studies on peptidic hydrogel prediction and de novo design are less explored and remain challenging[26,37].

This work provides an integrated computational, experimental, and machine learning (ML) approach to build a score function for discovering tetrapeptides for hydrogelation with an improved hit rate. Tetrapeptides have sufficient structural and sequence diversity for

developing a peptide hydrogel library with ample candidates, while requiring a moderate workload of simulation for generating training data. This approach proceeds as follows, firstly, the computation adopts CGMD and ML-trained regression model to provide an estimation of AP (Fig. 1a). Based on the original score function $AP_H$[33], 55 peptides are selected and chemically synthesized (Fig. 1b) for verification of gelation. With the resulting gelation feasibility (*i.e.*, yes or no), a classification model is trained to produce the gelation corrector $C_g$ fed to the original score function. An updated score function is then devised as $AP_{HC}$ (Fig. 1b). The process above is looped three times with mutual information exchange between ML and experimental results (Fig. 1b) to enhance the performance of $C_g$ from experimental results

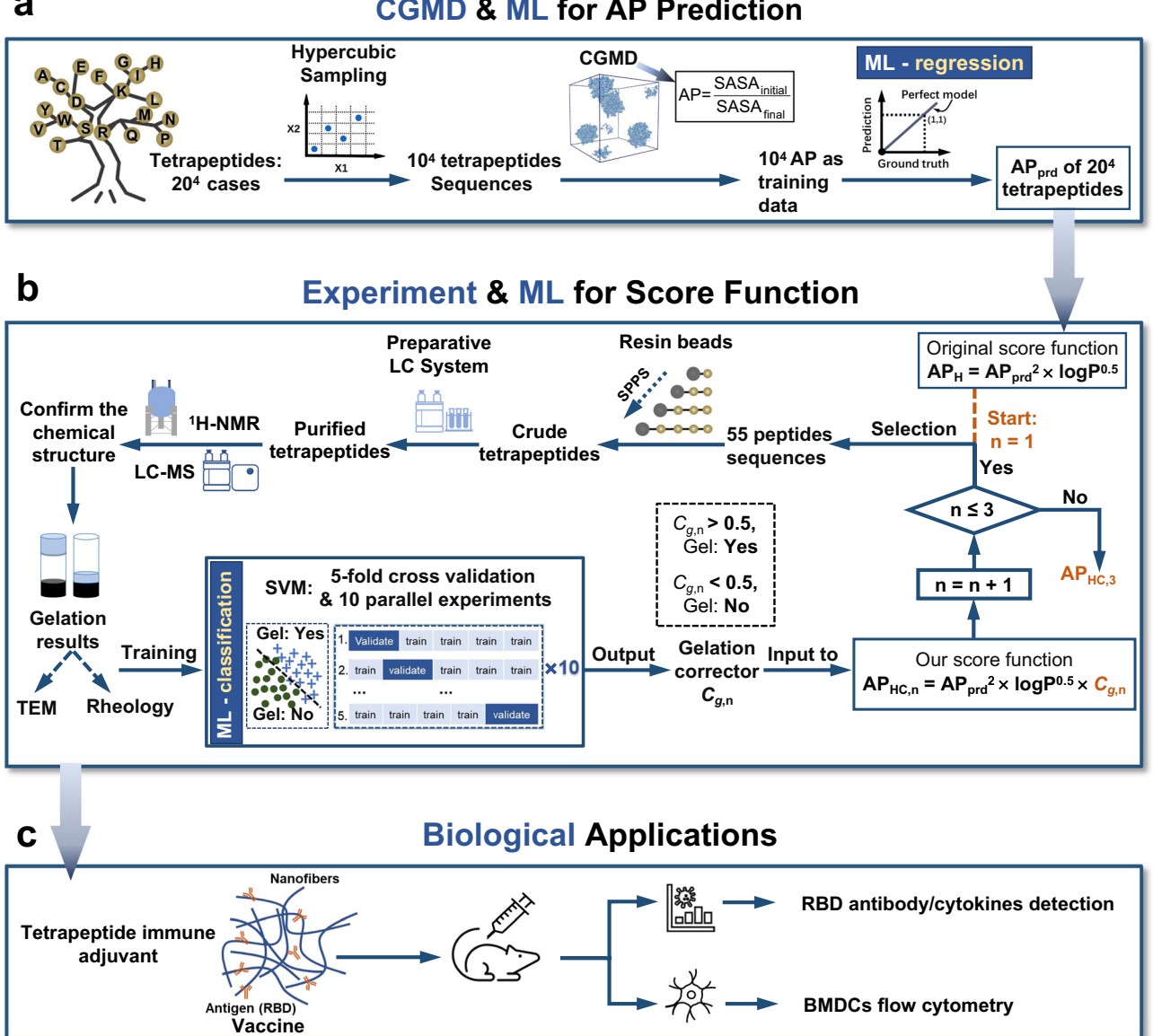

## a   CGMD & ML for AP Prediction

Hypercubic Sampling

CGMD

$$AP = \frac{SASA_{initial}}{SASA_{final}}$$

ML - regression

Tetrapeptides: $20^4$ cases → $10^4$ tetrapeptides Sequences → $10^4$ AP as training data → $AP_{prd}$ of $20^4$ tetrapeptides

## b   Experiment & ML for Score Function

Preparative LC System

Resin beads

Original score function $AP_H = AP_{prd}^2 \times logP^{0.5}$

Confirm the chemical structure

$^1$H-NMR

LC-MS

Purified tetrapeptides ← Crude tetrapeptides ← SPPS ← 55 peptides sequences ← Selection

Start: n = 1

Yes

$n \leq 3$   No

$AP_{HC,3}$

n = n + 1

$C_{g,n} > 0.5$, Gel: Yes

$C_{g,n} < 0.5$, Gel: No

Gelation results

Training

ML - classification

SVM:   5-fold cross validation & 10 parallel experiments

Gel: Yes
1. Validate | train | train | train | train
2. train | validate | train | train | train
... ... ×10
5. train | train | train | train | validate
Gel: No

Output → Gelation corrector $C_{g,n}$ → Input to → Our score function $AP_{HC,n} = AP_{prd}^2 \times logP^{0.5} \times C_{g,n}$

TEM   Rheology

## c   Biological Applications

Nanofibers

Tetrapeptide immune adjuvant

Antigen (RBD)
Vaccine

RBD antibody/cytokines detection

BMDCs flow cytometry

**Fig. 1 | Workflow of coupled experimental and machine learning approach for discovering tetrapeptide hydrogels and their potential biological applications. a** $10^4$ uniformly distributed tetrapeptide sequences are obtained by hypercubic sampling first. CGMD simulations are then performed to generate the training data of aggregation propensity (AP), and regression models are trained to predict the AP of the entire sequence space of tetrapeptides ($20^4$). **b** Based on the available score function $AP_H = AP^2 \times logP^{0.5}$, 55 peptides are selected and chemically synthesized to verify the gelation ability (gel marked with 1 and non-gel marked with 0). The sequence (feature) and the 1/0 (label) data are then passed to the ML algorithm to train a classification model, producing a gelation corrector $C_g$ for each tetrapeptide. The $AP_H$ score is then updated to $AP_{HC,1} = AP^2 \times logP^{0.5} \times C_{g,1}$, and another batch containing 55 peptides is selected based on $AP_{HC}$ and is synthesized and verified. Then, the whole 110 sequences (feature) and 1/0 (label) data are employed to update the classification model to generate $C_{g,2}$, and the $AP_{HC,1}$ score are updated to $AP_{HC,2} = AP^2 \times logP^{0.5} \times C_{g,2}$. Based on $AP_{HC,2}$, the third batch of 55 peptides are selected and chemically synthesized, and $C_{g,2}$ and $AP_{HC,2}$ are updated to $C_{g,3}$ and $AP_{HC,3}$. **c** The de novo designed peptide hydrogel is applied to serve as an efficient adjuvant for enhancing antibody production of RBD protein.

of 165 peptides, 100 of which could form hydrogels after gelation tests. Finally, tetrapeptide hydrogels obtained by de novo design from our computational model are selected as immune adjuvants to boost humoral immune recognition towards the receptor binding domain (RBD) of SARS-CoV-2 virus (Fig. 1c). The results show that the selected tetrapeptide hydrogel boosts the immune response of a model protein RBD from the spike protein of coronavirus. Overall, an 8,000-peptide library for gelation is built based on $AP_{HC}$ with a gelation rate reaching 87.1% (Supplementary Data 2), providing great potential for further innovations in peptide-based soft materials.

## Results

### Performance of corrected score function $AP_{HC}$

We employed cost-effective ML prediction instead of performing brute-force CGMD for generating the AP values of the entire space of tetrapeptides containing 160,000 sequences. Therefore, accurate prediction of AP values relying on ML regression models should be a prerequisite for obtaining potential hydrogels. We tested various training conditions, including training algorithms, feature representation approaches, and the size of training datasets to obtain an optimal AI model (Supplementary Figs. 2–4, Supplementary Tables 2–4, and Supplementary Data 1). Using the algorithm of support vector machine (SVM)[38] with 10,000 training data represented by 80-bit one-hot approach with amino acid sequence (Supplementary Table 3), we obtained a reliable SVM model with training/testing performance of 0.095/0.092 in mean absolute difference ($MAE_{tr}/MAE_{te}$) and 0.928/0.933 in coefficient of determination ($R^2_{tr}/R^2_{te}$)[39] (Fig. 2a, b). Further analysis of the prediction performance of SVM model revealed that the error between the predicted AP ($AP_{prd}$) and simulated AP ($AP_{sim}$) was less than 2.5% as $AP_{sim}$ was greater than 1.5 (Fig. 2c), proving the reliability and capability of the selected model on predicting peptide aggregates and further formation of hydrogels.

Distinctive from all available score functions focusing on the prediction of peptide self-assembly[33], we constructed a corrected score function $AP_{HC}$ within three loops (Fig. 2d–f) for improving the gelation hit rate. Since the final goal was to develop a hydrogel-forming peptide library with the minimum candidate numbers and the highest gelation possibility, we constrained our gelation hit rate assessment within the top 8000 assessing scores ($AP_H$ and $AP_{HC}$). We calculated $AP_H$ (Fig. 1b) in the first loop and randomly selected 55 peptides (26 peptides that were among the top 8000 in the $AP_H$ ranking), which were possibly to form hydrogel according to human expertise. It was found that 16 among the 26 peptides (within the top 8000) could form a hydrogel, and a corresponding gelation hit rate of 61.5% could be achieved with the $AP_H$ score, while a similar hit rate of 63% can be achieved with the $AP_{prd}$ score alone (Fig. 2d, left panel). With the total 55 gelation results, we trained a classification model to generate the gelation corrector $C_{g,1}$ with an averaged accuracy of 0.735 (averaged over ten parallel ML experiments, Fig. 2d, right panel). During the second loop, we calculated $AP_{HC,1}$ ($AP_{HC,1} = AP_{prd}^2 \times logP^{0.5} \times C_{g,1}$) and selected another 55 peptides, 30 of which were in the top 8000 $AP_{HC,1}$, and 23 peptides (of the 30 peptides) formed hydrogels, resulting in a hit rate of 76.7% within the top 8000 $AP_{HC,1}$ pool, while the $AP_H$ score yielded only a gelation hit rate of 64.7% (Fig. 2e, left panel). Augmenting the gelation results from the second batch to the first batch (total 110 data), we retrained a classification model to update the gelation corrector from $C_{g,1}$ to $C_{g,2}$ with an average accuracy of 0.746 (Fig. 2e, right panel). Proceeding to the third loop, we updated the $AP_{HC,1}$ to $AP_{HC,2}$ ($AP_{HC2} = AP_{prd}^2 \times logP^{0.5} \times C_{g,2}$). Similar to the previous two loops, we selected 55 peptides, and a gelation hit rate of 81.6% (31 out of 38) was generated within the top 8000 $AP_{HC,2}$ and a rate of 75.0% was achieved with $AP_{HC,1}$ alone (Fig. 2f, left panel). With a total of 165 experimental gelation results, a final classification model was trained and produced gelation corrector $C_{g,3}$ with an averaged accuracy of 0.767 (Fig. 2f, right panel) and score $AP_{HC,3}$

($AP_{HC,3} = AP_{HC} = AP_{prd}^2 \times logP^{0.5} \times C_{g,3}$) for each peptide, and a gelation hit rate of 87.1% was finally achieved with the top 8,000 $AP_{HC,3}$ (Fig. 2g), while the $AP_{prd}$ and $AP_H$ could only produce a gelation hit rate around 66% (Fig. 2h) based on the 165 gelation results. We listed the top 8,000 $C_g$ and $AP_{HC}$ (Supplementary Data 2 and 3) peptides for the convenience of selection and comparison.

To further differentiate between $AP_{HC}$ and $AP_H$ in predicting peptide hydrogels, we next compared the relationship between $AP_{HC}$ and logP' (Fig. 2i) as well as $AP_H$ and logP' (Fig. 2j) of experimentally synthesized 165 peptides that were marked with blue (gelation: yes) or red (gelation: no) dots, and those of total tetrapeptides (gray dots). Here, logP' indicated normalized hydrophilicity between 0 and 1. In addition to the relationship between $AP_{HC}$ and logP', the relationship between $AP_{HC}$-AP and $AP_{HC}$-$C_g$ was also investigated (Supplementary Fig. 6a, b). No linear correlation for $AP_{HC}$ and logP' (also $AP_{HC}$ and AP) can be observed, demonstrating that hydrophobicity and aggregation propensity were not the only two contributors to gelation, for instance, lower isoelectric points (i.e., 4.5 - 6 on pH scale) could improve the gelation performance (Supplementary Fig. 6c) due to the Columbic interaction and hydrogen bonds, inducing the formation of water-containing networks between deprotonated peptides and water solvent. These results indicated the significance of cooperating experimental input (i.e., $C_g$) into a prediction of hydrogel-forming sequences. Furthermore, it was conducive for hydrogelation when logP' was in the range of 0.05 to 0.4, as evidenced that the logP' of all gelating peptides were in this range (Fig. 2i). Peptides with too weak hydrophilicity (<0.05) possibly form precipitates while ones with too strong hydrophilicity (>0.4) maintain in solution. The $AP_H$ also assigned high scores to peptides with logP' in the range of 0.05 to 0.4. However, $AP_H$ cannot efficiently pinpoint peptides with high gelation potential and low $AP_{prd}$. As a result, more gelation peptides fall out of top 8000 compared to $AP_{HC}$ (Fig. 2j). We have also compared the ranks of $AP_{HC}$ and $AP_H$ of the complete sequence space of tetrapeptides (Fig. 2k). $AP_{HC}$ can significantly increase the rank of peptides which could potentially form hydrogels (maximum absolute difference in rank between $AP_{HC}$ and $AP_H$ is $8.5 \times 10^4$), such as WVII (by 14311) and IMVV (by 57146), while decreasing the rank of peptides that hardly form the hydrogel, such as WPYY (by 33033) and WWCP (by 50677). These four peptides were synthesized, validating that WVII and IMVV can form hydrogel while WPYY and WWCP cannot (Fig. 2l, Supplementary Data 6 and 7).

### Discovery and characterization of peptide hydrogels

After validating the efficiency of $AP_{HC}$ in predicting tetrapeptide hydrogels, we detailed the phase state of 165 synthesized tetrapeptides and the observed assembly behavior in an aqueous solution. Having demonstrated the identity of each synthetic tetrapeptides by mass spectrometry (MS, Supplementary Data 4) and nuclear magnetic resonance spectroscopy (NMR, Supplementary Data 5), we defined the hydrogel as the formation of a self-supporting, non-flowing mixture of water and hydrogelator through the vial-inverting method. Figure 3a (Insert optical images) showed the 6 representative tetrapeptides (FVIY, WEFF, WKFF, WTIF, WVFY, and IFYT) hydrogels in the glass vial, probably due to the π-π interaction of more than two aromatic amino acids in the tetrapeptide. Transmission electron microscope (TEM) studies (Fig. 3a and Supplementary Data 6) showed that the hydrogel formed by FVIY, WEFF, WKFF, or WTIF contained entangled nanofibers, while the hydrogel formed by WVFY or IFYT contained interlaced nanosheets. MD simulations (1250 ns) confirmed the observation of TEM results, and the front ranking of $AP_{HC}$ demonstrated the formation of these hydrogels. Mechanical properties of tetrapeptide hydrogels (Fig. 3b and Supplementary Fig. 7) indicated that both the elasticity (G') and the viscosity (G'') exhibited weak frequency dependence between 0.01 and 100 Hz. The G' values were higher than G'' values, suggesting the formation of a hydrogel.

 

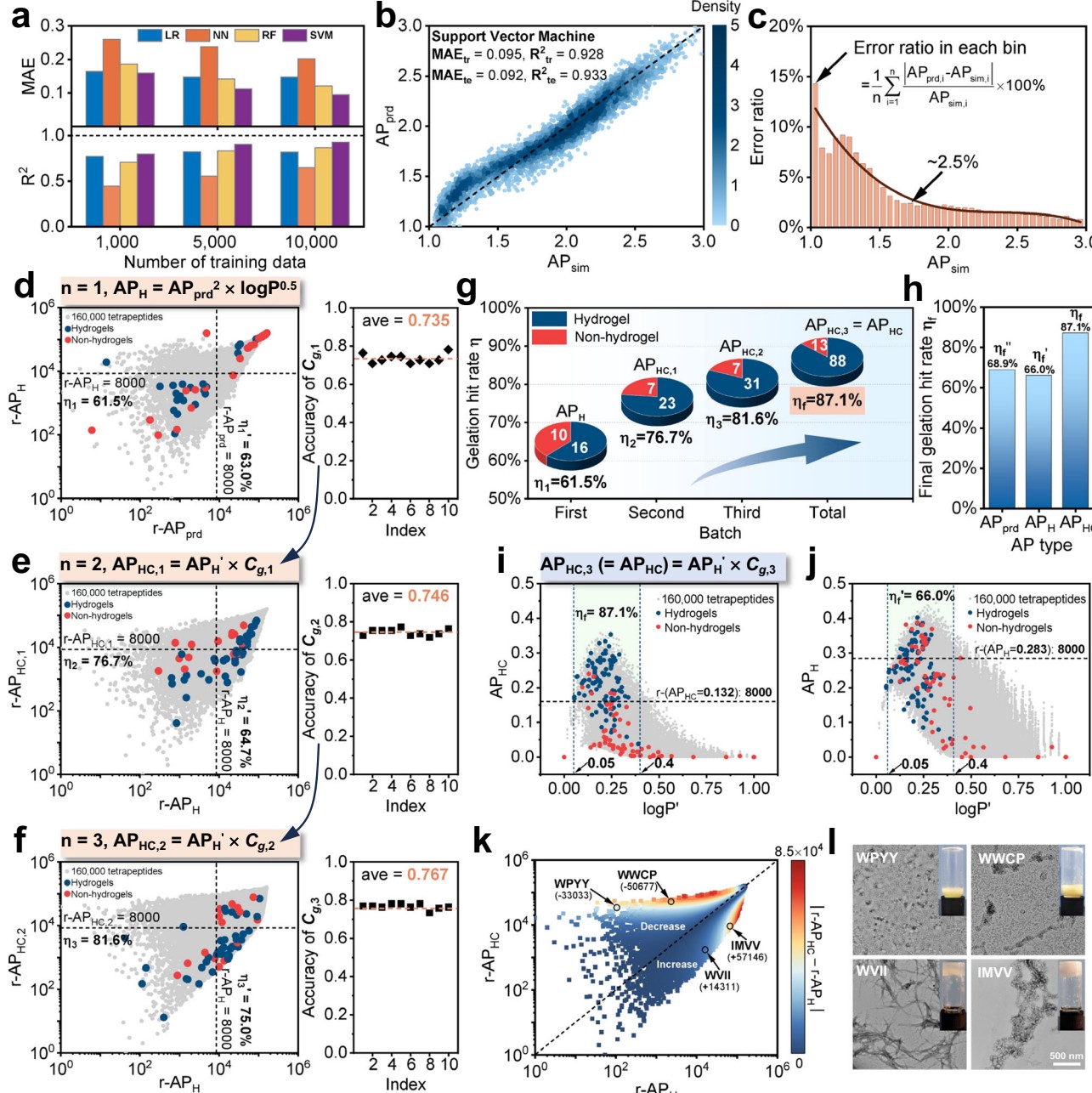

**Fig. 2 | "Human-in-the-loop" for obtaining corrected score AP$_{HC}$. a** Performance of different algorithms (*i.e.*, LR: linear regression; NN: nearest neighbor; RF: random forest; SVM: support vector machine) with different numbers of training datasets (*i.e.*, 1,000, 5,000, and 10,000). **b** Training and testing performance of ML model trained with SVM and 10,000 data using one-hot representation. The color scale indicates the density of the data points. **c** Error distribution with respect to simulated AP value (AP$_{sim}$). **d** First batch: ranking of experimentally selected peptides with respect to AP$_{prd}$ and AP$_H$, and accuracy of resulted $C_{g,1}$. The Chi-square statistic test (single-sided test) has been performed, with the null hypothesis that the proportion of hydrogel-forming peptides in the population of top 8000 AP$_H$ score is larger or equal to 61.5%, with a degree of freedom of 1 and significance level of 0.05. **e** Second batch: ranking of experimentally selected peptides with respect to AP$_H$ and AP$_{HC,1}$, and accuracy of results in $C_{g,2}$. The Chi-square statistic test (single-sided test) has been performed with a degree of freedom of 1 and a significance level of 0.05. **f** Third batch: ranking of experimentally selected peptides with respect to AP$_H$

and AP$_{HC,2}$, and accuracy of resulted $C_{g,3}$. The Chi-square statistic test (single-sided test) has been performed with a degree of freedom of 1 and a significance level of 0.05. **g** Gelation hit rate of experimentally synthesized tetrapeptides within the top 8000 ranking with respect to AP$_H$ (first batch), AP$_{HC,1}$ (second batch), AP$_{HC,2}$ (third batch), and AP$_{HC,3}$ (final). **h** The comparison between final gelation hit rates evaluated by AP$_{prd}$, AP$_H$, and AP$_{HC}$. **i** (The Chi-square statistic test (single-sided test) has been performed with a degree of freedom of 1 and significance level of 0.05.) and **j** Distribution of AP$_{HC}$ and AP$_H$ with respect to logP' of experimentally synthesized 165 tetrapeptides (blue indicates gelation while red indicates solution) and the complete sequence space of tetrapeptides (gray). **k** Comparison between the ranking of AP$_{HC}$ (r-AP$_{HC}$) and AP$_H$ (r-AP$_H$), where color indicates the absolute difference between r-AP$_{HC}$ and r-AP$_H$ of a single tetrapeptide. **l** TEM images of WPYY, WWCP, WVII, and IMVV (Inserts: optical images of the corresponding peptide). Source data are provided as a Source Data file.

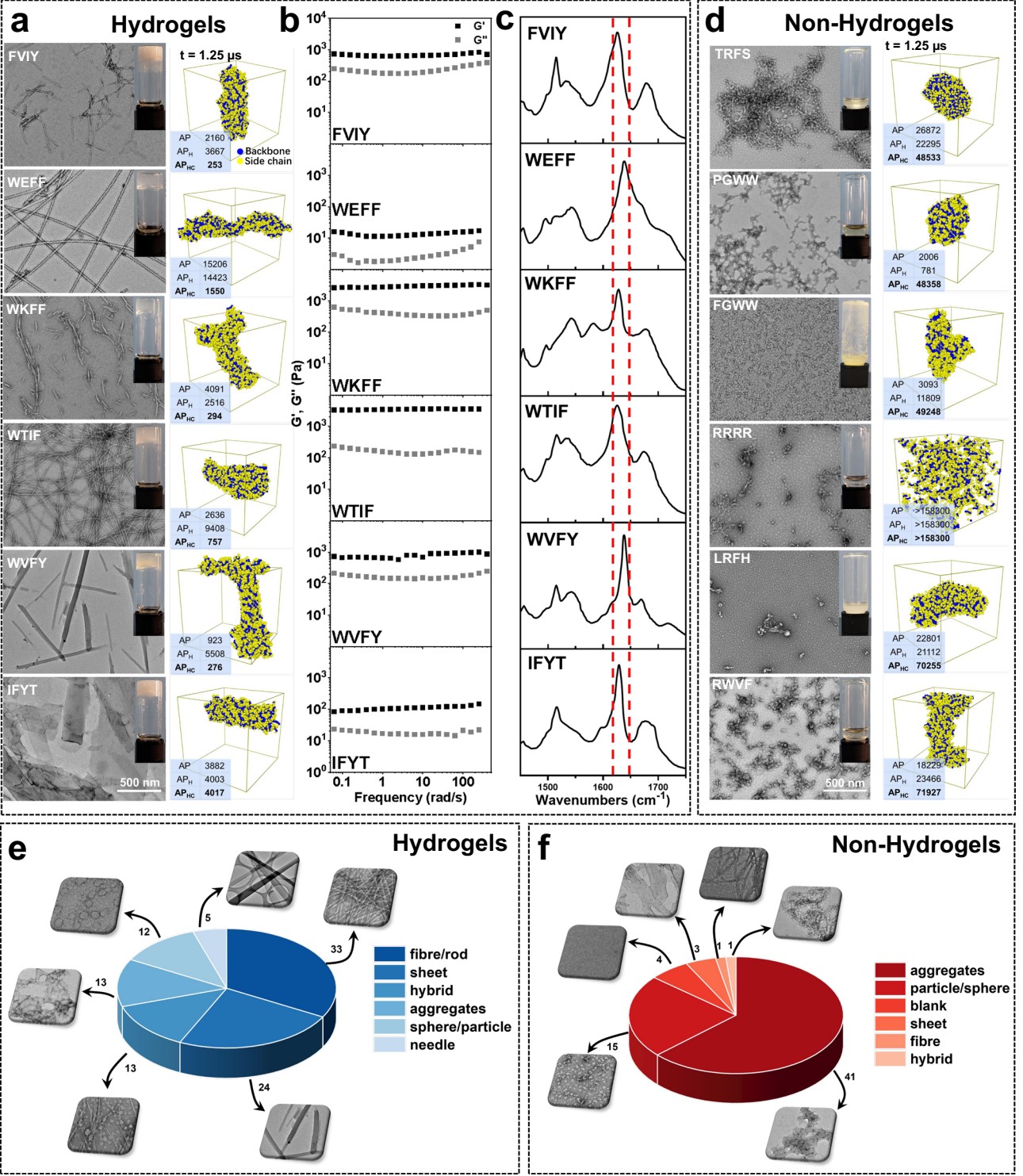

**Fig. 3 | Experimental investigations on the self-assembly behavior of 165 synthetic tetrapeptides. a** TEM images of 6 representative hydrogels of synthetic tetrapeptides, respectively. Inserts: optical images of the corresponding hydrogel (pH between 7.0 to 7.5). MD simulation results (1250 ns) and $AP_{HC}$ ranking were shown in the right column. **b** Dynamic frequency sweep of tetrapeptide hydrogels at the strain value of 0.5%. **c** FTIR spectra in the amide I region of tetrapeptide hydrogels. **d** TEM images of 6 representative non-hydrogels of tetrapeptide Inserts: optical images of corresponding solution/suspension (pH = 7.5). MD simulation results (1250 ns) and $AP_{HC}$ ranking were shown in the right column. **e** Statistics and classification of morphologies obtained by TEM for hydrogel-forming tetrapeptides (100 peptides). **f** Statistics and classification of morphologies obtained by TEM for non-hydrogel-forming tetrapeptides (65 peptides). Source data are provided as a Source Data file.

Fourier transforms infrared (FTIR) spectroscopy (Fig. 3c) in the amide I region (1620–1648 cm$^{-1}$, C = O stretching vibration) revealed the presence of β-sheet conformation in all these six hydrogels, further indicating the presence of highly ordered peptide nanostructures.

We also paid attention to those non-hydrogel-forming tetrapeptides to obtain rules of sequences of non-gelating peptides. Figure 3d (Insert optical images) showed six representative tetrapeptides (TRFS, PGWW, FGWW, RRRR, LRFH, and RWVF) with low $AP_{HC}$ ranking, some

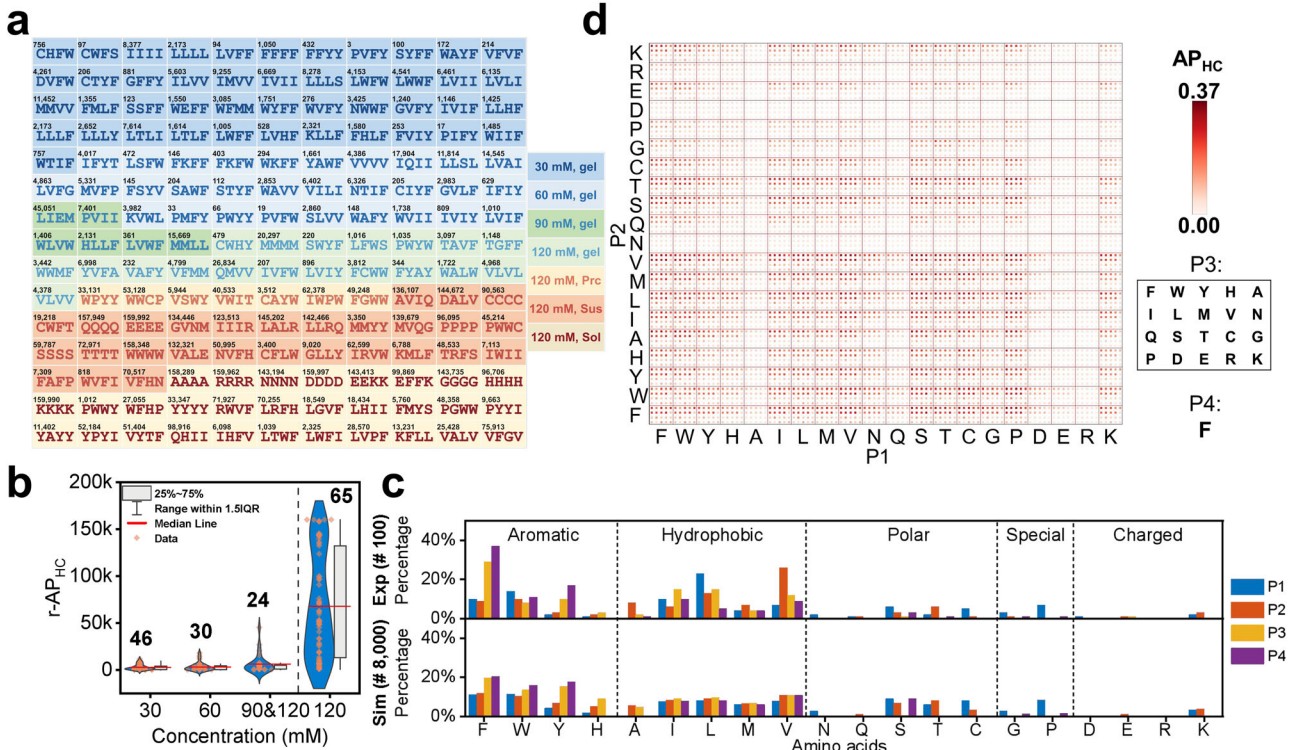

**Fig. 4 | Hydrogelation laws from experiments and simulations. a** Sequences of 165 synthesized tetrapeptides. The number in each fill represents the rank of $AP_{HC}$. Different color represents the hydrogelation capability at different concentrations. **b** Averaged rank of $AP_{HC}$ of hydrogel-forming peptides at 30, 60, 90, and 120 mM, as well as non-gelating peptides at 120 mM (after the dashed line), $n = 46$ for 30 mM, $n = 30$ for 60 mM, $n = 24$ for 90 and 120 mM (with gelation), and $n = 65$ for 120 mM (without gelation, n represent the number of synthetic tetrapeptides in each category. **c** Contribution of each amino acid at different positions to hydrogel formation, compared between 100 experimental data of hydrogel-forming peptides and the top 8,000 $AP_{HC}$ simulation data. **d** Distribution of 8,000 $AP_{HC}$ with amino acid F fixed at the C terminus (P4). The x-axis is P1 (N-terminus), the y-axis is P2, and the third position is illustrated in the rectangular box. Source data are provided as a Source Data file.

of which were highly soluble while others were barely soluble in water. TEM images (Fig. 3d and Supplementary Data 6) showed that these six peptides formed aggregates with different sizes in an aqueous solution, which were qualitatively consistent with the morphologies obtained in MD simulations except for RRRR (Fig. 3d, right column), showing different levels of aggregation. Taking the TEM result of RRRR together, we attributed this to the thermodynamic factor of concentration. Finally, we presented a summary of the assembled morphologies of all synthesized tetrapeptides (Fig. 3e, f and Supplementary Data 6), indicating that hydrogel-forming tetrapeptides tended to form fibers, sheets, or hybrid morphology (70%) in an aqueous solution. Non-hydrogel-forming tetrapeptides self-assembled into aggregates, spheres, or particulate supramolecular structures (86%). The results above confirmed the self-assembled nanostructures and hydrogelation results of these synthetic tetrapeptides, as predicted by the corrected score function $AP_{HC}$.

### Hydrogelation laws from experiment and simulation results

One hundred and sixty-five synthesized peptides were presented with different colors indicating the capabilities of hydrogel formation (Fig. 4a). The average rank of $AP_{HC}$ (Fig. 4b) for peptides gelation at four certain concentrations was 2664, 2801, 3646, and 4899, respectively, which was consistent with the experimental results (Supplementary Data 7) of the gelation capability, demonstrating the reliability of $AP_{HC}$ in screening tetrapeptides for the hydrogel formation.

Hydrogelation laws (i.e., the effect of position and type of amino acids on gelation) deduced from the experimentally synthesized peptides gelators (100 data) and computationally selected candidates (top 8000 data based on $AP_{HC}$) exhibited reasonable consistency

(Fig. 4c). Aromatic amino acids (F and Y) had the largest contribution to gelation, especially when located at positions 3 and 4 near the C-terminus. The W had a much lower contribution due to the strong hydrophobicity, which may lead to suspension with precipitation instead of forming a hydrogel. The H amino acid with a five-membered ring structure was favored in position 3 in gelation peptides. Second to F, W, and Y, the amino acids I, L, V, and M can also contribute to gelation due to hydrophobicity carried by side chains. The simulation results only slightly increased the percentage of I, L, V, and M at positions 2 and 3, while the experimental results raised the percentage of I, L at positions 1 and 3 and V at position 2. The contribution of the polar amino acids N, Q, S, T, and C to gelation was identical in both the experiment and simulation. N and Q with strong polarity were rarely found in gelation peptides with scarce occupancy at position 1 or 2. S and T with moderate polarity were beneficial for gelation when S was located at positions 1, 2, and 4 and T at 1, 2. Apolar amino acid C contains the -SH group, which may induce the formation of disulfide bonds and stable nanostructures, especially when located at position 1. Amino acid P contributed to the hydrogel formation when located at position 1 because of the potential formation of the "kink" structure[33,40], promoting self-assembly. Meanwhile, G without functional side chains cannot significantly contribute to gelation. Charged amino acids D, E, R, and K had a minimal contribution to gelation. However, peptides with K near the N-terminus were found to form hydrogel due to the attraction of opposite charges driving self-assembly.

To gain an overview of the effects of position-type on gelation, we analyzed the $AP_{HC}$ scores of the complete space sequence of tetrapeptides with fixed position 4 (fixed F, Fig. 4d; fixed remaining

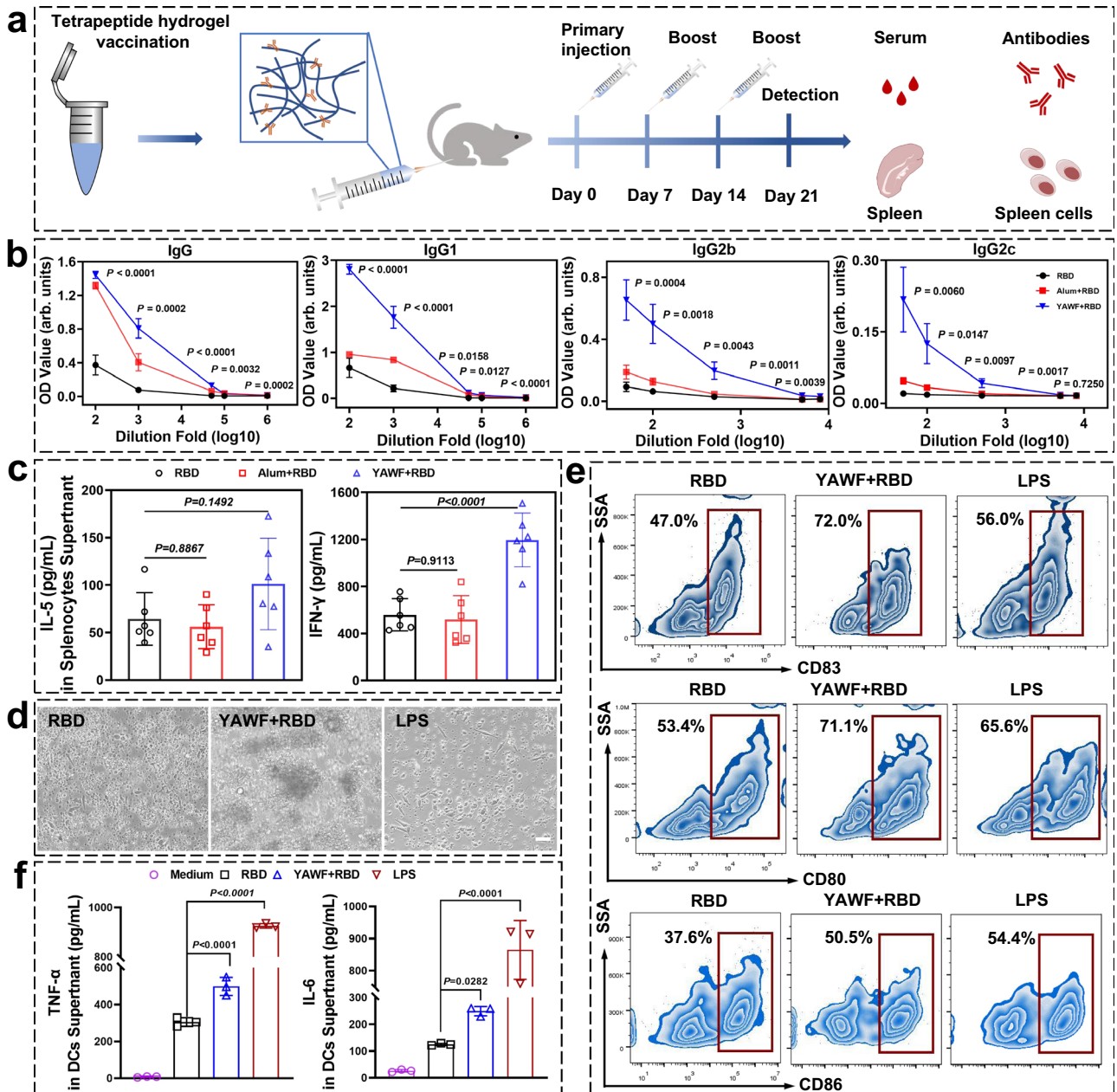

**Fig. 5 | Response to tetrapeptide-based hydrogel nano vaccination. a** 6–8 weeks C57BL/6 mice were immunized thrice at day 0, 7, and 14 with 15 μg RBD (RBD group), 12.5 μL aluminum adjuvant, and 15 μg RBD (Alum + RBD group), 60 mM tetrapeptide hydrogel and 15 μg RBD (YAWF + RBD group). Serum and splenocytes were collected on day 21. **b** Enzyme-linked immunosorbent assay (ELISA) responses to serum samples (RBD-specific) at different dilutions. SARS-CoV-2 RBD-specific IgG antibodies (IgG, IgG1, IgG2b, and IgG2c) were analyzed by endpoint dilution ELISA and measured as absorbance at 450 nm. The data were shown as the mean ± SEM (*n* = 6 biologically independent mice), and the p values were calculated by comparing RBD with YAWF + RBD using a one-way ANOVA test. **c** 7 days after the last immunization, splenocytes were collected and re-stimulated with RBD protein.

The bars shown were mean ± SD (*n* = 6 biologically independent samples), and differences between RBD and other treatments were determined using a one-way ANOVA test. The secretion of IL-5 and IFN-γ in the splenocytes supernatants was detected using ELISA. **d** Optical images of bone marrow-derived dendritic cells (BMDCs) treated with RBD-loaded tetrapeptide hydrogel (scale bar = 100 μm). **e** Flow cytometry analysis of BMDCs expressing CD83, CD80, and CD86. **f** The level of IL-6 and TNF-α in BMDCs culture supernatants were analyzed using ELISA. The data were shown as the mean ± SD (*n* = 3 biologically independent samples), and differences between RBD and other treatments were determined using one-way ANOVA test. Source data are provided as a Source Data file.

19 amino acids, Supplementary Figs. 8–26). Different from Fig. 4c, we can discern the effect of doublets and triplets of amino acids on gelation, other than a single amino acid. It can be confirmed that aromatic-aromatic (F, W, Y – F, W, Y) and aromatic-hydrophobic (F, W, Y – I, L, M, V) doublets had positive effects on gelation synergistically. In addition, aromatic amino acids bonded with P and K exhibited similar positive performance. These rules can also be applied to the triplets. In addition, we analyzed the position-type percentage with

respect to adjacent amino acids, based on the 100 hydrogel-forming peptides in the experiment and 8000 peptides with the highest APHC score in the simulation (Supplementary Fig. 27). It can also be deduced that aromatic-aromatic and aromatic-hydrophobic doublets have the most significant contribution to hydrogelation, and position-specific rules regarding other amino acids are also congruent with those deduced from Fig. 4c, d. For example, amino acid A is barely found in the fourth position, except when F or Y is located in the third position.

In summary, we have presented a complete picture of the relationship between the gelation ability and position & type of 20 natural amino acids, providing schematic guidance for experimentalists to design tetrapeptide hydrogels and possibly functional applications associated.

## Boosting antibody production of RBD vaccine

The advantage of self-assembling peptide materials is their remarkable multivalency, which contributes to improved immunogenicity. It is generally known that multivalency can repeatedly display ligands or epitopes to increase affinity for specific receptors while enhancing antibody responses[8,9,41–43]. The RBD of the spike protein covering the surface of SARS-CoV-2 attracted our interest as a promising target antigen for COVID-19 vaccines[44–47]. We hypothesized that tetrapeptide hydrogel could provide a biodegradable platform to encapsulate RBD protein and enhance humoral immune responses against RBD protein. Since YAWF has a $AP_{HC}$ rank within the top 8000 (1661, Fig. 4a) and can form hydrogels containing nanofibrous network (Supplementary Fig. 28), we selected this tetrapeptide as a vaccine adjuvant candidate. We quantified the production of antigen-specific antibodies in C57BL/6 mice, which was a crucial indicator for evaluating the performance of the SARS-CoV-2 vaccine (Fig. 5a). Compared with the RBD group, the results (Fig. 5b) showed that the FDA (U.S. Food and Drug Administration) approved adjuvant aluminum could enhance the generation of IgG by 20.7-fold. The hydrogel formed by YAWF remarkably increased the generation of IgG by 41.6-fold (the endpoint titres of RBD, aluminum, and YAWF were shown in Supplementary Fig. 29), suggesting that the tetrapeptide hydrogel could boost the immune response in vivo. The results also indicated that the hydrogel group significantly enhanced the production of IgG1, which was similar to the aluminum group. The RBD-specific IgG2b response in the hydrogel group increased around 9.7-fold, compared with the commercial aluminum adjuvant group. As for IgG2c, the hydrogel formed by YAWF maintained high IgG2c titers, surpassing the ones in the aluminum group or control group (Fig. 5b and Supplementary Fig. 29).

During the infection, the regular pathway to produce IgG antibody is highly related to the proliferation of SARS-CoV-2-specific CD8⁺ or CD4⁺ T cells, which is reflected by the elevated secretion of several typical cytokines, including interleukin-5 (IL-5) and interferon-γ (IFN-γ). Compared with the aluminum adjuvant group, the mice that received YAWF based vaccine showed a higher IL-5 level in their splenocytes culture, and IFN-γ secretion was also obviously evoked (Fig. 5c). Thus, the YAWF stimulated an obvious cell-dependent adaptive immune response. To further confirm the capability of the tetrapeptide vaccine to regulate related cell immunity, the upstream dendritic cells (DCs) activation enhanced by tetrapeptide hydrogel was evaluated. The DCs treated with YAWF vaccine showed promising activation as the percentage of CD83, CD80, and CD86 expressing cells augmented to 72.0%, 71.1%, and 50.5% (Fig. 5e and Supplementary Fig. 30). Such intense activation could also be proved by the clustering of DCs (Fig. 5d) producing raised levels of Th-1 cytokines (Fig. 5f). To sum up, de novo designed tetrapeptide hydrogels as immune adjuvant successfully enhanced the immune response to RBD protein in vivo, providing great inspiration for us to explore natural tetrapeptide hydrogel library for biomedical applications.

## Discussion

This work demonstrated an efficient "human-in-the-loop" framework that integrated coarse-grained molecular dynamics, machine learning, and experiments for the prediction and discovery of peptide hydrogels. The framework evolved into an updated score function $AP_{HC}$ to evaluate the hydrogelation feasibility of 160,000 natural tetrapeptides, and a gelation hit rate of 87.1% with the top 8,000 $AP_{HC}$ rank was achieved. The simulation and experiment revealed similar hydrogelation laws for short peptide design. Subsequently, a de novo-designed

tetrapeptide hydrogel based on our hydrogelation laws was successfully applied in SARS-CoV-2 vaccine adjuvant, proving the potentials of the peptide libraries within the top 8,000 $AP_{HC}$ rank for developing versatile biological and medical applications. Moving forward, the "human-in-the-loop" framework can be further automated by employing a robotic platform for synthesizing new peptides and performing machine learning for training classification models. The framework described here can also be extended to the efficient design of other functional materials/devices, including the terminal-covered peptide hydrogels, peptide batteries, peptide fluorescence probes, and peptide semiconductors, contributing to modern organic nanotechnology employing short peptide building blocks as key structural and functional elements.

## Methods

### Ethical approval

All mice were handled in accordance with institutional guidelines, and all animal experiments were approved by the Institutional Animal Care and Use Committee (IACUC) of Westlake University (IACUC Protocol #21-046-WHM).

### Material sources

Fmoc-amino acids were obtained from GL Biochem (Shanghai, China). 2-Cl-trityl chloride resin was obtained from Nankai Resin Co. Ltd. (Tianjin, China). Commercially available reagents were used without further purification unless noted otherwise. Deionized water was used for all experiments. All other chemicals were reagent grade or better. Horseradish peroxidase-conjugated goat anti-mouse IgG, IgG1, IgG2b, and IgG2c (1030-05, 1071-05, 1091-05, and 1078-05, 1:5000 dilution) were obtained from Southern Biotech (USA). Mouse IFN-γ, IL-5, TNF-α and IL-6 ELISA kits (430807, 431204, 430907, and 431307) were obtained from Biolegend (USA). Recombinant murine GM-CSF and IL-4 (315-03 and 214-14) were purchased from Peprotech (USA). FITC anti-mouse CD83 Antibody, FITC anti-mouse CD80 Antibody, and PE anti-mouse CD86 Antibody (121505, 104705, and 159204, 0.5 μg per million cells in 100 μL volume for usage) were purchased from Biolegend (USA). Imject Alum Adjuvant was obtained from Thermofisher (USA). RBD-Fc (Z03513, SARS-CoV-2 Spike protein, RBD, mFc Tag, CHO-expressed) and RBD-H (Z03479, SARS-CoV-2 Spike protein, RBD, His Tag,) were purchased from Genscript Biotech (Nanjing, China). This research followed institutional guidelines, and all animal experiments were approved by the Institutional Animal Care and Use Committee of Westlake University.

### Coarse-grained molecular dynamics (CGMD)

To speed up the simulation/screening, coarse-grained molecular dynamics (CGMD) simulations were adopted to generate machine learning (ML) training data, which were performed with the open-source GROMACS package[48,49] and Martini2 force field[50–52]. The all-atom tetrapeptide structures (prepared based on CHARMM36[53]) were coarse-grained using the python script martinize.py[51]. In simulations for screening purposes, total of 300 coarse-grained tetrapeptides (as zwitterions) were solvated randomly in a 13 nm × 13 nm × 13 nm box with water whose density was set as approximately 1 g/cm³ (-18700 water beads). The charge of the tetrapeptide/water system was maintained neutral by adding the proper amount of Na⁺ or Cl⁻, and the system was also maintained at neutral pH. The whole system was then energy-minimized using the steepest descent algorithm[54], until the maximum force on each atom was less than 20 kJ mol⁻¹ nm⁻¹. Subsequently, the system was passed to an equilibration run for $5 × 10^6$ steps with a time step of 25 fs, resulting in a total simulation time of 125 ns. The temperature and pressure during the equilibration were controlled through Berendsen algorithm at 300 K and 1 bar, respectively. A total of 15,000 such simulations were performed, and the selection of the initial 15,000 tetrapeptides was based on Latin hypercubic

sampling[55]. To obtain more accurate and stable morphology of self-assembled structure, a 1,250 ns duration was employed, and the final morphology results were averaged over 8 identical simulations.

To quantitatively characterize the degree of self-assembly, we adopted the aggregation propensity (AP) value[33], which was calculated by:

$$AP = \frac{SASA_{initial}}{SASA_{final}} \qquad (1)$$

Where the $SASA_{initial}$ and $SASA_{final}$ are the solvent (*i.e.*, tetrapeptides) accessible surface area at the beginning and end of a CGMD equilibration run.

Self-assembled peptides cannot guarantee the formation of hydrogels, which was also affected by the hydrophobicity of peptides. Therefore, a hydrogel formation score function $AP_{HC}$ considering hydrophobicity was utilized to screen out the peptides with the highest possibility of hydrogel formation under current experimental/computational conditions, as shown below:

$$AP_{HC} = AP' \times logP' \times C_g \qquad (2)$$

$$logP = \sum_{i=1}^{4} \triangle G_{wat-oct,i} \qquad (3)$$

Where the AP' and the logP' are normalized AP and logP value (normalized to 1), and $\alpha$ (=2) and $\beta$ (=0.5) are two coefficients determining the significance of AP' and the logP'. $\Delta G_{wat-oct,\,i}$ (kcal mol$^{-1}$) is the Wimley–White whole-residue hydrophobicities for each amino acid[56]. $C_g$ is the gelation corrector output by the ML classification model trained with experimental gelation results.

## Machine learning

Four different ML algorithms were deployed: Random Forest (RF)[57], Linear Regression (LR)[58], Nearest Neighbor (NN)[59], and Support Vector Machine (SVM)[60]. Mean absolute error (MAE) and coefficient of determination (R$^2$)[39] were calculated to assess the performance of each ML model. Different numbers of training data sets prepared by Excel 2019 (i.e., 1000, 5000, and 10,000) were used to train the ML models. In each training, 80% of the training data was used for training, while the remaining 20% was used for validation (Fig. 1b). After obtaining each model, another 5000 data were employed for independent testing.

Before training the ML model, we converted the amino acid sequence into numerical data with 4-integer and 80 bit one-hot representation approaches (shown in Supplementary Table 3, taking Glu-His-Asn-Thr, i.e., EHNT, as an example), aiming for enhanced model performance with optimal data presentation approach. Moreover, a tetrapeptide can be considered as a "tripeptide" with each "position" represented by one of the 400 possible dipeptide sequences, namely, a tetrapeptide can have 1200 possible bits with 3 of them to be 1. Therefore, we also trained models with a 1200-bit one-hot representation converted from the dipeptide sequence composition.

All the model training and prediction were conducted via ASCENDS code[61], which employs a Python-based open-source data analytic toolkit, scikit-learn[62]. The training was initially performed based on the default hyperparameter settings in ASCENDS (Supplementary Table 5). To investigate the effect of hyperparameters on training performance, we selected three kernels and different parameters and ranges for tuning (Supplementary Table 6)[63]. The performance of the SVM model with varying hyperparameters were illustrated in Supplementary Fig. 5. The highest training performance of MAE$_{tr}$ was 0.090 and R$^2_{tr}$ was 0.934, as kernel = rbf, $C$ = 100, and gamma = 0.001. However, it was only slightly increased compared to

the generated training performance (MAE$_{tr}$ = 0.095, R$^2_{tr}$ = 0.928, as shown in Fig. 2b) with the default hyperparameters (kernel = rbf, $C$ = 1, and gamma = auto, equaling to 1/n_features = 1/80 = 0.0125). The slightly increased training performance would have minimal impact on the prediction, and we thus concluded that the default hyperparameters were good enough for achieving reliable prediction results.

## Synthesis, purification, and characterization of tetrapeptides

The selected tetrapeptides were synthesized by solid-phase peptide synthesis (SPPS) using 2-chlorotrityl chloride resin, and the side chains of the corresponding N-Fmoc protected amino acids were properly protected by different chemical groups (Supplementary Fig. 1). First, the C-terminal of the first amino acid was conjugated to the resin. Anhydrous N, N'-dimethyl formamide (DMF) containing 20% piperidine was used to remove Fmoc group. To couple the next amino acid to the free amino group, HBTU (O-(Benzotriazol-1-yl)-N, N, N', N'-tetramethyluronium hexafluorophosphate) was used as coupling reagent and the organic base N, N-diisopropylethylamine (DIPEA) was added. The growth of the peptide chain was performed according to the established Fmoc SPPS protocol. After the final coupling step, the excess reagent was rinsed with DMF, followed by five washing steps using dichloromethane (DCM) for 1 min (5 mL per gram of resin). The peptide was cleaved using cleavage reagent (trifluoroacetic acid (TFA): triisopropylsilane (TIS): H$_2$O = 95%: 2.5%: 2.5) for 45 minutes. 20 mL per gram of resin of ice-cold diethyl ether was then added to the concentrated cleavage reagent. The resulting precipitate was centrifuged at 1500 g for 10 minutes at 4 °C. The supernatant was then decanted, and the resulting solid was dissolved in H$_2$O/CH$_3$CN (1:1) for HPLC separation. HPLC was conducted at Agilent 1260 Infinity II Manual Preparative Liquid Chromatography system using a C18 RP column with CH$_3$CN (0.1% of trifluoroacetic acid) and water (0.1% of trifluoroacetic acid) as the eluents (Supplementary Table 1). The purity of each tetrapeptide was verified by HPLC, and the purified tetrapeptide was dissolved in 200 μL of 0.1 mg/mL methanol to prepare Mass spectrometry (MS) samples. MS was conducted at the Agilent InfinityLab LC/MSD system with MSD signal set as positive ion mode. NMR samples were prepared by dissolving purified tetrapeptides in 600 μL of 8 mg/mL DMSO-d6. $^1$H NMR spectra were obtained on a Bruker BioSpin AVANCE NEO spectrometer (500 MHz, Switzerland), using tetramethyl silane as an internal standard. The structure of tetrapeptide was verified by MS and $^1$H NMR.

## Transmission electron microscope

We used the negative staining technique to observe the morphologies formed by peptides. A micropipette was used to load 10 μL of sample solution to a carbon-coated copper grid, and we used a piece of filter paper to remove the excess solution. After rinsing the grid with the deionized water, we used uranyl acetate to stain the sample for 1 minute and then rinsed the grid with deionized water again. The excess liquid was drained with filter paper and conducted on a Talos L120C system, operating at 120 kV.

## Peptide hydrogel formation

This work defines hydrogel formation as a self-supporting, non-flowing mixture of water and hydrogelator by the vial-inverting method. All purified tetrapeptides were dissolved in ultrapure water (to form a 30 mM solution initially), followed by the stepwise addition of 1 N NaOH solution to adjust the overall aqueous pH to 6.5–8.5. Meanwhile, a short-term ultrasonication treatment was applied after each pH adjustment to facilitate peptide solubilization and speed up peptide self-assembly. These operations could be repeated several times until a viscous, translucent colloid was formed, suggesting the initial stage of the gelation process. The mixture was allowed to stand in for another 48 h for complete hydrogelation. Upon the absence of gelation phenomenon during the aforementioned loops, we increased the peptide

concentration (60 mM, 90 mM, and 120 mM were selected) and re-started such gelation operation loop to explore their gelation feasibility.

## Rheology

During the experiment, a rheology test was carried out on an ARES-G2 (TA instrument) system with a 25 mm parallel plate at a 500 μm gap during the experiment. In the process of dynamic frequency (strain = 0.5%) scanning, the obtained hydrogel was transferred to the test platform with a pipette, and the changes of elastic modulus (G′) and viscous modulus (G″) of the hydrogel during scanning (frequency from 100–0.01 Hz) was tested. The hydrogels were then characterized by dynamic strain sweep at a fixed frequency of 1 Hz, and the changes in elastic modulus (G′) and viscous modulus (G″) of the hydrogel were recorded (strain: 0.01–100%).

## Fourier transform infrared spectroscopy

Samples of tetrapeptide hydrogel were first lyophilized to powder, then we placed the sample on a diamond single reflection attenuated total reflectance (ATR) module. Spectra were recorded on a FTIR micro spectrometer (ThermoFisher Nicolet iS50) by averaging 32 scans at a spectral resolution of 1 cm$^{-1}$.

## Preparation of tetrapeptide hydrogel vaccines

YAWF was dissolved in 600 μL of endotoxin-free PBS buffer (≤0.5 EU/mL, Cellcook Biotech Co. Ltd., Guangzhou, China). A homogeneous hydrogel was formed by adjusting the final pH to 7.5 by 1 N NaOH. Then RBD protein (RBD-Fc) was added to the hydrogel, followed by vortexing and standing at room temperature for one hour to obtain a tetrapeptide hydrogel protein vaccine. For in vivo immune evaluation, C57BL/6 J mice were randomly divided into three groups (n = 6): (1) 15 μg RBD protein (RBD group); (2) 12.5 μL aluminum adjuvant and 15 μg RBD protein (Alum + RBD group); (3) 60 mM tetrapeptide hydrogel and 15 μg RBD protein (YAWF + RBD group). Each mouse was injected subcutaneously in the groin with 100 μL of the prepared vaccine.

## Mice

C57BL/6 J mice (female, 6-8 weeks old) were obtained from the laboratory animal resources center (LARC) at Westlake University. They were housed in specific pathogen-free (SPF) conditions, and a 12-h light/12-h dark cycle was used. The housing temperature for mice is between 20–26 °C with 40–70% humidity.

## Tetrapeptide hydrogel in promoting dendritic cells maturation

Bone marrow cells were isolated from the femur and tibia of C57BL/6 J mice and then cultured in 1640 medium containing GM-CSF (5 ng/mL) and IL-4 (5 ng/mL) at 37 °C for 6 days[64]. The collected immature DCs were plated in a 24-well plate at a density of $1 \times 10^6$ cells per well. After 24 h, 50 μL of the blank medium, vaccine, and LPS were added to each well, respectively, then the medium volume was supplemented to 1 mL. The cells were cultured for another 24 h and centrifuged to collect the cells and supernatant. The acquired cells were labeled with FITC-tagged anti-CD80, FITC-tagged anti-CD83, and PE-tagged anti-CD86 for flow cytometry (CytExpert Software for CytoFLEX 2.4.0.28). The production of IL-6 and TNF-α in the cell culture supernatants was also analyzed by ELISA kit.

## ELISA for antibody titer

The production of anti-RBD IgG, IgG1, IgG2b, and IgG2c antibodies in mice serum was analyzed by ELISA. RBD proteins (RBD-H) were plated on 96-well uncoated ELISA plates (Biolegend) at 3 μg/mL in PBS buffer overnight at 4 °C. The plate was blocked with 1% BSA for 2 h at 37 °C. By washing the plate three times with PBST (Phosphate buffer saline (PBS) with 5‰ Tween 20), 100 μL diluted mice serum was added into per well

and incubated at 37 °C for 2 h. The plate was washed with PBST four times. Each well of 96-well plate was added with 100 μL HRP-labeled goat anti-mouse IgG, IgG1, IgG2b, and IgG2c binding antibody (1: 5000 diluted in blocking buffer) at 37 °C for 1 h. After washing the plate four times with PBST, 50 μL 3,3′,5,5′-tetramethylbenzidine (TMB) was added into per well. The reaction was stopped with 50 μL 2 M $H_2SO_4$. The absorbance value at 450 nm and 570 nm wavelength was determined by a Microplate reader (Thermo Fisher Scientific, Varioskan LUX). Titers were analyzed with log 10 serum dilution plotted against absorbance at 450 nm minus absorbance at 570 nm. Antibody titer values were defined as the highest serum dilution that gave an optical density above 0.1.

## Cytokine production

On the 7th day after the last immunization, fresh splenocytes were collected by grinding the mouse spleen[65]. The splenocytes ($5 \times 10^6$ cells/mL) from each group of mice were plated in 24-well plates, and stimulated with soluble RBD protein (50 μg/mL) for 96 h. The production of IFN-γ and IL-5 in cell culture supernatants was analyzed by ELISA kit.

## Statistics and reproducibility

Statistical significance was determined using a one-way ANOVA test. Statistical analyses were performed using GraphPad Prism 8. For all representative TEM or optical images, experiments were performed three times independently with similar results.

## Reporting summary

Further information on research design is available in the Nature Portfolio Reporting Summary linked to this article.

## Data availability

All relevant data supporting the key findings of this study are available within the article and its Supplementary information files. Any additional requests for information can be directed to, and will be fulfilled by, the lead contact. Source data are provided with this paper.

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

## Acknowledgements

T.X., D.C., and H.W. are supported by the National Natural Science Foundation of China (82022038), and the Research Center for Industries of the Future (RCIF) at Westlake University. J.W., S.Z., and W.L. are supported by RCIF under Award No. WU2022C041. T.X. acknowledges the support of the Zhejiang Postdoctoral Science Foundation (No. 102216582101). J.W. also acknowledges the support of the Zhejiang Postdoctoral Science Foundation (No. 103346582102) and the National Natural Science Foundation of China (No. 52101023). This research was supported by Instrumentation and Service Centers for Molecular Science and for Physical Science, respectively, as well as by Biomedical Research Core Facilities at Westlake University.

## Author contributions

T.X. and J.W.: Conceptualization, Visualization, Methodology, Formal analysis, Computational simulation, Writing-original draft. S.Z. and D.C.: Formal analysis, Computational simulation, Writing-review & editing. H.Z., Y.F., N.K., and Z.Z.: Methodology, Resources, Writing-review & editing. W.L. and H.W.: Conceptualization, Supervision, Funding acquisition, Writing-review & editing. All the authors discussed and commented on the manuscript.

## Competing interests

The authors declare no competing interests.
