## [Peer review file · Nature Communications]

REVIEWER COMMENTS

Reviewer #1 (Remarks to the Author):

The authors have developed an iterative approach for robust prediction and design of peptide hydrogels that involves the mutual exchange between experiment and prediction. In the first step, they chemically synthesized more than 160 natural tetrapeptides and identified their ability to form hydrogels. They then employed experimental iterative loops to improve the accuracy of the gelation prediction. Finally, peptides derived from the prediction are selected as immune adjuvants to enhance immune recognition of SARA-Cov2. This work is interdisciplinary research and has the potential to be published in this publication. The following comments need to be addressed:

In order to develop machine learning models, the authors used one-hot encoding as an input feature. The logic behind it is understandable since it is a tetrapeptide. However, it would be better if the authors developed the model using amino acid and dipeptide composition encoding, and compared its performance with one-hot encoding. They can also experiment with hybrid features and see whether they improve prediction accuracy.

For training the machine, the authors used the entire dataset. The authors should use 80% of the data for training and 20% for independent validation. The comparison of performance between training and independent validation will provide insight into the generalization ability of the model. Afterwards, the authors can make a blind prediction.

In the introduction, it would be better to discuss why tetrapeptide has greater importance than other peptide compositions. I believe that this information will be useful for a general audience.

Parameter optimization is one of the most important aspects of machine learning. They did not provide any information regarding the parameter search range and employed cross-validation techniques to optimize the parameter. In the supplementary information, please provide the optimal parameters for each algorithm.

The SVM performance in Figure S3 is missing from the supplementary information. Please fix it.

Reviewer #2 (Remarks to the Author):

The manuscript by Xu and colleagues details approaches to improve the predictability of the behaviour of self-assembling peptide hydrogels. I limit my review to the immunological aspects of the paper. Here the authors tested the ability of tetrapeptide hydrogels to augment immunogenicity of the SARS-CoV-2 RBD, demonstrating improvements in humoral and potentially cellular immunity compared to no adjuvant or alum. Overall, the case that the YAWF hydrogel is acting as an immune adjuvant has been made. I was unclear how this data dovetails with the rest of the paper, since there was no prediction of adjuvant activity nor deployment of algorithmic approaches to improve this activity.

Major Comments:

1 There is limited explanation of the immunisation model selected. Why subcutaneous dosing, why 3 doses so quickly together, why 15ug? For comparison, human COVID vaccines were given intramuscularly 3-4 weeks apart, with immunogenicity generally assessed 2weeks after final dose.

2 The ELISA data appear inconsistent. The alum and hydrogel groups appear to elicit comparable levels of total IgG, but alum elicited markedly less of each IgG subclass. Where is the total IgG signal for the alum animals coming from then?

3 Line 303 – “aluminum could enhance the generation of IgG by 20.7-fold. The hydrogel formed by YAWF remarkably increased the generation of IgG by 41.6-fold”. Where are these numbers coming from, how were they derived? I assume these are endpoint titres?

4 Line 316 - “Thus, the YAWF stimulated intense T cell dependent adaptive immune response.” This is not supported by the data and the language should be moderated. The authors measured cytokines in bulk splenocyte cultures. As such, the source of the cytokines cannot be automatically assigned to T cells.

5 Line 319 – “The DCs treated with YAWF vaccine showed promising activation as the percentage of CD83, CD80, CD86 expressing cell augmented to 72.0%, 71.1% and 50.5% (Fig. 5e).” – The DC cultures treated with RBD protein alone showed significant activation too, as well as robust TNF and IL-6 production. Why is this the case in presumably unvaccinated animals, where addition of a simple protein should be relatively inert in culture. Gating for the medium only controls should be provided for comparison.

6 – Groups were compared using parametric t tests, which are not appropriate for small animal studies (N=6) or tissue culture experiments (N=3) where normalcy of the data cannot be established.

Reviewer #3 (Remarks to the Author):

The manuscript by Wang et al. demonstrated an integrated approach to building a score function for predicting and discovering tetrapeptide hydrogels. The approach combined coarse-grained molecular dynamics, machine learning and experiments, which built a reliable score function APHC through mutual information exchange between machine learning and experimental gelation feasibility. A remarkable gelation hit rate of 87.1% was achieved. Furthermore, as an application of the hydrogelation laws discovered, a tetrapeptide hydrogel successfully boosted the immune response of the RBD of SARS-CoV-2 in a mice model as a COVID-19 vaccine adjuvant. This work reveals may lead to further understanding self-assembly induced hydrogelation. The combine of simulation and experiments to formulate a design principle for hydrogel would greatly accelerate the biomedical applications of peptide hydrogels. Therefore, I recommend acceptance of this manuscript to Nature Communications after addressing the following issues.

1. It would be more comprehensive if the authors explain why the gelation hit rate within the top 8000 APHC rank was selected as a crucial parameter when evaluating the performance of the score function.
2. In Figure 4c, S is more common at positions 1, 2 and 4, while the authors claimed that it is beneficial for gelation when S is located at positions 1, 2 and 3. Please explain this.
3. The authors are encouraged to discuss the advantages of investigating tetrapeptides for hydrogelation. How about tripeptides or pentapeptides or even polypeptides?
4. If the C- and N- terminals are covered with functional groups or motifs, could the hydrogelation performance be possible to be predicted?
5. I found the title could be more precise. What is the meaning of “human-in-the-loop”, experiment data or supervised learning?
6. How does APHC correlate with the properties of tetrapeptides, for example, isoelectric points?
7. What are the pH values of the hydrogels formed? I would suggest the authors add that information in the captions of proper figures.
8. Is it surprising that both QQQQ and EEEE form suspension at 120 mM? Any explanation from either ML perspective or thermodynamic prospective?

The following are our responses to the comments (in Italics) of the reviewer and the changes (underlined) in the manuscripts.

A) Reviewer 1

[The authors have developed an iterative approach for robust prediction and design of peptide hydrogels that involves the mutual exchange between experiment and prediction. In the first step, they chemically synthesized more than 160 natural tetrapeptides and identified their ability to form hydrogen. They then employed experimental iterative loops to improve the accuracy of the gelation prediction. Finally, peptides derived from the prediction are selected as immune adjuvants to enhance immune recognition of SARA-Cov2. This work is interdisciplinary research and have the potential to be published in this publication.]

We are grateful for the insightful comments of the reviewer, and we have addressed all of the concerns of the reviewer.

The following comments need to be addressed:

[1. In order to develop machine learning models, the authors used one-hot encoding as an input feature. The logic behind it is understandable since it is a tetrapeptide. However, it would be better if the authors developed the model using amino acid and dipeptide composition encoding, and compared its performance with one-hot encoding. They can also experiment with hybrid features and see whether they improve prediction accuracy.]

We thank the reviewer for the suggestion. We discerned possible discrepancy between the reviewer's understanding of the "one-hot" representation and our actual encoding approach. In fact, the one-hot encoding in our original manuscript precisely captures the amino-acid composition in a tetrapeptides. Specifically, an amino acid is represented by 20 bits within only one "1" bit (there are 20 types of amino acids) A tetrapeptide, having four amino-acid units, is then represented by 80 ($= 20 \times 4$) bits (*i.e.*, features), as illustrated in **Table S3**. The 80-bit feature and label (*i.e.*, AP) is then fed to different algorithms for training ML models. In our original work, a tetrapeptide is also attempted to be represented by 4 decimal integers, each varying from 1 to 20. However, since this approach introduces discretization and hierarchies, the model generally exhibits poorer performance than that of the one-hot representation. Thus, the 4-integer representation is not adopted for model training.

As suggested by the reviewer, we have also trained ML models using dipeptide composition and compared its performance with that of amino acid composition. It should be noted that the dipeptide composition is also presented with one-hot approach using 1,200 ($= 400 \times 3$) bits. A tetrapeptide can be taken as a "tripeptide" with each "position" represented by one of the 400 possible dipeptide sequences, namely, a tetrapeptide can have 1200 possible bits (*i.e.*, feature) with 3 of them to be 1. We tested three datasets with 1,000, 5,000, and 10,000 data, respectively, with two algorithms of random forest (RF) and support vector machine (SVM). In the training process, 80% of the data are used for training and remaining 20% are used for validation. After obtaining the ML model, we also employed another 5,000 data for testing the performance of the model, and the MAE and R^2 shown here are thus obtained both on training and testing, as shown in **Table S4**. It can be observed that the performance of one-hot presentation using dipeptide composition are generally poorer than the counterpart of that using amino acid composition (as shown in **Table**

S1). This can be attributed to that, for high-dimensional input data space distributing over a more complex manifold, more training data was required for maintaining or improving the performance of the models. Therefore, it was foreseeable that using hybrid feature would further increasing the dimensionality of input data space and thus decreasing the model performance with the same number of datasets.

Revisions in the Supplementary Information:

Tab. S3 Representation of amino acid sequence of a tetrapeptide (taking EHNT as an example) by two approaches. First row is the single-letter representation of 20 amino acids, second row is the corresponding integer for each amino acid, and fourth row is the 4-integer representation of tetrapeptide EHNT, while the fifth row is the 80-bit representation of EHNT with amino acid composition encoding.

A	C	D	E	F	G	H	I	K	L	M	N	P	Q	R	S	T	V	W	Y
1	2	3	4	5	6	7	8	9	10	11	12	13	14	15	16	17	18	19	20
Position 1				Position 2				Position 3				Position 4							
4				7				12				19							
000100000...00000000				000000100...00000000				000000000001...0000				0000...0000000000010							
↙ Forth bit				↙ Seventh bit				↗ Twelfth bit				↗ Nineteenth bit							

Tab. S4 Training and testing performance of four algorithms with three different number of training data and four algorithms. The training data is represented with one-hot approach using 1200-bit converted from dipeptide composition.

Peptide Representation	# Training data	Algorithm	MAE _{tr}	R ² _{tr}	MAE _{te}	R ² _{te}
1200-bit representation (one-hot representation)	1,000	LR	>100	< 0	-	-
		NN	0.318	0.216	0.280	0.374
		RF	0.280	0.384	0.275	0.380
		SVM	0.296	0.362	0.282	0.400
	5,000	LR	0.147	0.819	0.140	0.832
		NN	0.258	0.470	0.250	0.487
		RF	0.193	0.689	0.185	0.704
		SVM	0.184	0.742	0.176	0.764
	10,000	LR	0.132	0.855	0.129	0.858
		NN	0.250	0.501	0.242	0.522
		RF	0.168	0.756	0.161	0.769
		SVM	0.147	0.832	0.140	0.846

Revisions in the main text:

We tested various training conditions, including training algorithms, feature representation approaches, and the size of training datasets, to obtain an optimal AI model (Fig. S1-S3 and Tab. S1-S4). Using the algorithm of support vector machine (SVM)³⁸ with 10,000 training data

represented by 80-bit one-hot approach with amino acid sequence (Tab. S3), we obtained a reliable SVM model with training/testing performance of 0.095/0.092 in mean absolute difference (MAE_{tr}/MAE_{te}) and 0.928/0.933 in coefficient of determination (R²_{tr}/R²_{te})³⁹ (Fig. 2a and b). Further analysis of the prediction performance of SVM model revealed that the error between the predicted AP (AP_{prd}) and simulated AP (AP_{sim}) was smaller than 2.5% as AP_{sim} was larger than 1.5 (Fig. 2c), proving the reliability and capability of the selected model on predicting peptide aggregates and further formation of hydrogels.

Revisions in Method of “Machine Learning”:

Before training of the ML model, we converted the amino acid sequence into numerical data with two approaches (shown in Tab. S3, taking Glu-His-Asn-Thr, *i.e.*, EHNT, as an example), aiming for enhanced model performance with optimal data presentation approach. In the first approach, termed as 4-integer representation, we labeled the 20 amino acids as integers from 1 to 20, and a tetrapeptide sequence will then be a sequence of four integers. In the second approach termed as 80-bit one-hot representation, we turned the four positions of a tetrapeptide into binaries with 80 bits, with each position containing 20 bits, while only one bit being 1 denoting a specific amino acid in the position, while the resting 19 bits being 0, aiming to eliminate the bias introduced due to the hierarchy of integers from 1 to 20. In addition, a tetrapeptide can be taken as a “tripeptide” with each “position” represented by one of the 400 possible dipeptide sequences, namely, a tetrapeptide can have 1200 possible bits with 3 of them to be 1. Therefore, we also trained models with 1200-bit one-hot representation converted from dipeptide sequence composition.

[2. For training the machine, the authors used the entire dataset. The authors should use 80% of the data for training and 20% for independent validation. The comparison of performance between training and independent validation will provide insight into the generalization ability of the model. Afterwards, the authors can make a blind prediction.]

We thank the reviewer for this insightful comment. In a routine machine training process, validation dataset is normally unavoidable as a part of training for tuning parameters, and in our training process, we did divide each training dataset as 80% for training, and 20% for validation (as shown in original Figure 1b: ML-classification, although it is for ML classification, the regression process is the same). If that is the meaning of “independent validation”, then the original work has fulfilled the requirements of machine training. Here, we would prefer to interpret the “independent validation” as “independent testing”. In the original work, after obtaining the ML model, we employ another 4,997 (now we have added another 3 data for testing task, making the testing data 5,000 in total) data for independent testing, and a similar even better MAE_{te} and R²_{te} performance are achieved. To clearer show the performance of MEA and R² in both training and prediction, we have appended those results to **Table S1** in Supplementary Information. It should be noted that the testing performance is dependent on the number of testing datasets, and normally larger number of testing data yields better performance. The performance comparison between the training and testing can well justify the generalization of our ML model. Based on the SVM model with optimal training performance (MAE_{tr} = 0.095, and R²_{tr} = 0.928), we make blind predictions of AP within the complete sequence space of tetrapeptides (*i.e.*, totally 160,000 peptides).

Revisions in the Supplementary Information:

We have added both the training and testing MAE and R^2 to Fig. S1-S3 and updated the caption of Fig. S1-S3 accordingly.

Fig. S1 Training (MAE_{tr} , R^2_{tr}) and testing (MAE_{te} , R^2_{te}) performance with **1,000** training datasets with 80-bit representation, tested by 5,000 data.

Fig. S2 Training (MAE_{tr} , R^2_{tr}) and testing (MAE_{te} , R^2_{te}) performance with **5,000** training datasets

with 80-bit representation, tested by 5,000 data.

Fig. S3 Training (MAE_{tr} , R^2_{tr}) and testing (MAE_{te} , R^2_{te}) performance with **10,000** training datasets with 80-bit representation, tested by 5,000 data. It should be noted that, since the SVM model is the optimal one chosen for predicting the AP values of 160,000 tetrapeptides, the model performance is thus presented in main text as Fig. 2b, while not here for avoiding repeatability.

We deleted original Fig. S4-S6 since the 4-integer representation was redundant for this manuscript.

We have added the MAE_{te} and R^2_{te} in the Tab. S1 in supplementary materials. The caption of Tab. S1 is also changed accordingly.

Tab. S1 Training (MAE_{tr} and R^2_{tr}) and testing (MAE_{te} and R^2_{te}) performance of different ML algorithms and number of training data, with 80-bit data representation with amino acid composition and 4-integer representation of peptides. Since the 4-integer representation yields much worse training performance than the 80-bit approach, it was abandoned immediately and thus was not proceeded with testing. The training performance of MAE_{tr} and R^2_{tr} with 80-bit representation are averaged results over ten parallel ML experiment, shown in Tab. S2.

Peptide Representation	# Training data	Algorithm	MAE_{tr}	R^2_{tr}	MAE_{te}	R^2_{te}
80-bit representation	1,000	LR	0.155	0.804	0.154	0.800
		NN	0.255	0.513	0.242	0.543

(one-hot representation)		RF	0.186	0.728	0.173	0.755	
		SVM	0.158	0.804	0.152	0.819	
	5,000	LR	0.146	0.823	0.150	0.813	
		NN	0.227	0.579	0.219	0.597	
		RF	0.140	0.834	0.133	0.840	
		SVM	0.112	0.899	0.109	0.905	
	10,000	LR	0.147	0.821	0.150	0.813	
		NN	0.196	0.693	0.184	0.721	
		RF	0.119	0.871	0.113	0.881	
		SVM	0.095	0.928	0.092	0.933	
	4-integer representation	1,000	LR	0.332	0.142	-	-
			NN	0.319	0.186	-	-
RF			0.229	0.564	-	-	
SVM			0.306	0.240	-	-	
5,000		LR	0.339	0.147	-	-	
		NN	0.280	0.380	-	-	
		RF	0.192	0.701	-	-	
		SVM	0.290	0.357	-	-	
10,000		LR	0.336	0.147	-	-	
		NN	0.263	0.434	-	-	
		RF	0.168	0.764	-	-	
		SVM	0.281	0.382	-	-	

Since the MAE_{tr} and R^2_{tr} are averaged results over ten parallel ML experiments, we have added the MAE and R^2 in each ML experiment as Tab. S2 in supplementary materials and cited in main text at appropriate position.

Tab. S2 Training performance (MAE_{tr} and R^2_{tr}) of ten parallel ML experiments, trained based on three different number of datasets and four algorithms with 80-bit one-hot representation.

# Datasets	Algorithm	Index	MAE_{tr}	R^2_{tr}
1,000	LR	1	0.155	0.801
		2	0.160	0.786
		3	0.154	0.809
		4	0.153	0.809
		5	0.152	0.811
		6	0.153	0.810
		7	0.157	0.801
		8	0.154	0.805
		9	0.154	0.806
		10	0.155	0.806
	NN	1	0.258	0.502
		2	0.257	0.510

		3	0.253	0.522	
		4	0.256	0.517	
		5	0.255	0.519	
		6	0.262	0.484	
		7	0.255	0.524	
		8	0.252	0.517	
		9	0.251	0.530	
		10	0.255	0.505	
		RF	1	0.184	0.737
			2	0.187	0.726
	3		0.181	0.743	
	4		0.183	0.737	
	5		0.191	0.711	
	6		0.189	0.721	
	7		0.189	0.716	
	8		0.186	0.729	
	9		0.187	0.730	
	10		0.186	0.726	
	SVM	1	0.158	0.803	
		2	0.159	0.802	
		3	0.158	0.806	
		4	0.160	0.800	
		5	0.159	0.804	
		6	0.159	0.804	
		7	0.159	0.802	
		8	0.158	0.805	
		9	0.156	0.807	
		10	0.157	0.804	
	5,000	LR	1	0.145	0.824
			2	0.146	0.822
3			0.147	0.819	
4			0.146	0.823	
5			0.145	0.823	
6			0.145	0.823	
7			0.145	0.822	
8			0.145	0.823	
9			0.146	0.823	
10			0.145	0.823	
NN		1	0.226	0.583	
		2	0.227	0.579	

		3	0.228	0.573	
		4	0.226	0.581	
		5	0.228	0.573	
		6	0.225	0.585	
		7	0.228	0.573	
		8	0.226	0.583	
		9	0.226	0.582	
		10	0.227	0.576	
		RF	1	0.141	0.832
			2	0.141	0.833
	3		0.140	0.837	
	4		0.140	0.835	
	5		0.140	0.836	
	6		0.140	0.833	
	7		0.141	0.833	
	8		0.139	0.838	
	9		0.142	0.832	
	10		0.141	0.834	
	SVM	1	0.112	0.899	
		2	0.112	0.898	
		3	0.112	0.898	
		4	0.112	0.898	
		5	0.112	0.899	
		6	0.112	0.899	
		7	0.112	0.898	
		8	0.112	0.900	
		9	0.112	0.898	
		10	0.112	0.898	
	10,000	LR	1	0.147	0.821
			2	0.147	0.820
3			0.146	0.821	
4			0.147	0.821	
5			0.147	0.821	
6			0.146	0.821	
7			0.147	0.819	
8			0.147	0.821	
9			0.147	0.821	

		10	0.146	0.821
	NN	1	0.195	0.696
		2	0.197	0.690
		3	0.197	0.692
		4	0.197	0.688
		5	0.196	0.694
		6	0.195	0.696
		7	0.195	0.696
		8	0.195	0.696
		9	0.198	0.689
		10	0.197	0.692
	RF	1	0.119	0.872
		2	0.118	0.874
		3	0.119	0.871
		4	0.119	0.871
		5	0.119	0.871
		6	0.119	0.871
		7	0.119	0.872
		8	0.120	0.869
		9	0.119	0.871
		10	0.120	0.870
	SVM	1	0.095	0.928
		2	0.096	0.927
		3	0.095	0.928
		4	0.095	0.928
		5	0.095	0.929
		6	0.095	0.928
		7	0.096	0.928
		8	0.096	0.927
		9	0.095	0.928
		10	0.095	0.928

Revisions in the main text:

We have updated the Fig. 1b: Replaced “test” with “validate”.

We have added MAE_{te} and R^2_{te} to Fig. 2b and updated the caption of Fig. 2b accordingly.

Fig. 2 “Human-in-the-loop” for obtaining corrected score AP_{HC} . **a**) Performance of different algorithms (*i.e.*, LR: linear regression; NN: Nearest Neighbor; RF: random forest; SVM: Support vector machine) with different number of training datasets (*i.e.*, 1,000, 5,000, and 10,000). **b**) Training and testing performance of ML model trained with support vector machine and 10,000 data using one-hot representation. Color scale indicates the density of data points.

Revisions in Method of “Machine learning”:

Four different ML algorithms were deployed, they were Random Forest (RF)⁵⁸, Linear Regression (LR)⁵⁹, Nearest Neighbor (NN)⁶⁰, and Support Vector Machine (SVM)⁶¹, respectively. Mean absolute error (MAE) and determination coefficient (R^2)⁶² were calculated to assess the performance of each ML model. Different number of training data sets (*i.e.*, 1,000, 5,000, and 10,000) were used to train ML model for investigating the effect on performance. In each training, 80% of the training data were used for training, while the remaining 20% were used for validation (Fig. 1b). After obtaining each model, another 5,000 data were employed for independent testing.

[3. In the introduction, it would be better to discuss why terapeptide has greater importance than other peptide compositions. I believe that this information will be useful for a general audience.]

We thank the reviewer for the comment. In order to systematically design short peptide hydrogels, we use natural tetrapeptides as the basic material. Compared with natural dipeptides (400 peptides) and tripeptides (80,000 peptides), 160,000 natural tetrapeptides have sufficient structural and sequence diversity to develop a peptide hydrogel library with ample candidates, which has been widely studied in the field of biomedicine¹⁻⁴. In addition, their modest number of sequences also indicates reasonable computational power. Although pentapeptides (3,200,000 peptides) and even longer peptides have more diverse structures and sequence quantities, their computational simulations require much greater computing power for generating training data. To explore the self-assembling or hydrogelation of polypeptides or even proteins, more advanced ML or simulation technique should be employed/developed, which falls out of the scope of this research, aiming for developing a “Human-in-the-loop” scheme for enhancing the gelation hit rate.

Revisions in main text:

This work provides an integrated computation, experiment, and ML approach to build a score function for discovering tetrapeptides for hydrogelation with an improved hit rate. Tetrapeptides have sufficient structural and sequence diversity for developing a peptide hydrogel library with ample candidates, while requiring a moderate workload of simulation for generating training data. This approach proceeds as follows, firstly, the computation adopts CGMD and ML-trained regression model to provide an estimation of AP (Fig. 1a).

[4. Parameter optimization is one of the most important aspects of machine learning. They did not provide any information regarding the parameter search range and employed cross-validation techniques to optimize the parameter. In the supplementary information, please provide the optimal parameters for each algorithm.]

We thank the reviewer for the comment. In the training process, we used the default hyperparameters for training the model. The default hyperparameters for each algorithm are shown in **Table S5**, providing the necessary information for possible reproduction of any readers using ASCENDS⁵. Based on the default hyperparameters, we have achieved a SVM model with desirable training performance ($MAE_{tr} = 0.095$, and $R^2_{tr} = 0.928$ and the testing counterparts are $MAE_{te} = 0.092$, $R^2_{te} = 0.933$, as shown in **Table S1**). However, as the reviewer suggested, the default hyperparameters used probably are not the optimal ones, hence, we here investigated the effect of the hyperparameters on training performance with SVM algorithms using the 80-bit one-hot representation with 10,000 training data. In total, there are five types of kernels (“linear”, “poly”, “rbf”, “sigmoid”, and “precomputed”) in ASCENDS, among them, we chose three kernels (“linear”, “poly”, “rbf”) that are mostly used. For SVM, different hyperparameters and their range should be tuned according to the kernel and we only selected the most important parameters for tuning⁶ (shown as Table S6). The training performance with each kernel is shown in Fig. S4. The achieved greatest training performance of MAE_{tr} is 0.090 and R^2_{tr} is 0.934, as kernel = rbf, $C = 100$, and $\gamma = 0.001$, however, it is only slightly increased compared with the generated training performance ($MAE_{tr} = 0.095$, $R^2_{tr} = 0.928$, as shown in **Table S1**) of default hyperparameters (kernel = rbf, $C = 1$, and $\gamma = \text{auto}$, equaling to $1/n_{\text{features}} = 1/80 = 0.0125$). The slightly increased training performance will have minimal effect on the prediction, and we

thus conclude that the default hyperparameters are good enough for achieving reliable prediction results.

Revisions in the Supplementary Information:

Tab. S5 Default hyperparameter values for each algorithm.

Algorithm	Hyperparameters	Values
LR	fit_intercept	True
	positive	False
	copy_X	True
	n_jobs	None
RF	scaler_option	StandardScaler
	n_estimators	100
	max_features	Auto
	max_depth	None
	min_samples_split	2
	min_samples_leaf	1
	bootstrap	True
	criterion	Mse
	min_weight_fraction_leaf	0
	max_leaf_nodes	None
	min_impurity_decrease	0
NN	n_neighbors	5
	weights	Uniform
	algorithm	Auto
	leaf_size	30
	p	2
	metric	Minkowski
	metric_params	None
SVM	kernel	rbf
	degree	3
	coef0	0.0
	tol	1e-3
	C	1.0
	epsilon	0.1

	shrinking	True
	gamma	auto

Tab. S6 Kernels and associated parameters and their tuning range for SVM.

Kernel	Parameter and range
linear	C (0.0001, 0.001, 0.01, 0.1, 1, 10, 100, 1000)
poly	C (0.001, 0.01, 0.1, 1, 10, 100, 1000); degree (1, 2, 3, 4)
rbf	C (0.001, 0.01, 0.1, 1, 10, 100, 1000); gamma (0.001, 0.01, 0.1, 1)

Fig. S4 Performance of SVM model with different kernels and hyperparameters, trained with 10,000 data represented by 80-bit one-hot representation.

Revisions in Method of “Machine learning”:

All the model training and prediction were conducted via ASCENDS code⁶³, which employs a Python-based open-source data analytic toolkit, scikit-learn⁶⁴. The training was initially performed based on the default hyperparameters setting in ASCENDS (Tab. S5). To investigate the effect of hyperparameters on training performance, we selected three kernels and different parameters and ranges for tuning (Tab. S6)⁶⁵. The performance of the SVM model with varying hyperparameters were illustrated in Fig. S4. The achieved greatest training performance of MAE_{tr} was 0.090 and

R^2_{tr} was 0.934, as kernel = rbf, C = 100, and gamma = 0.001. However, it was only slightly increased compared with the generated training performance ($MAE_{tr} = 0.095$, $R^2_{tr} = 0.928$, as shown in Fig. 2b) with default hyperparameters (kernel = rbf, C = 1, and gamma = auto, equaling to $1/n_{features} = 1/80 = 0.0125$). The slightly increased training performance would have minimal effect on the prediction, and we thus concluded that the default hyperparameters were good enough for achieving reliable prediction results.

[5. The SVM performance in Figure S3 is missing from the supplementary information. Please fix it.]

We thank the reviewer for the comment. The SVM performance was shown as Fig. 2b, since it was the optimal model that we have trained and used in the prediction of AP. To avoid redundancy, we did not show it in Figure S3. Following the suggestion of the reviewer, we have added a notion in the caption of Fig. S3.

Revisions in the Supplementary Information:

Fig. S3 Training (MAE_{tr} , R^2_{tr}) and testing (MAE_{te} , R^2_{te}) performance with **10,000** training datasets with 80-bit representation, tested by 5,000 data. It should be noted that, since the SVM model is the optimal one chosen for predicting the AP values of 160,000 tetrapeptides, the model performance is thus presented in main text as Fig. 2b, while not here for avoiding repeatability.

B) Reviewer 2

[The manuscript by Xu and colleagues details approaches to improve the predictability of the behaviour of self-assembling peptide hydrogels. I limit my review to the immunological aspects of the paper. Here the authors tested the ability of tetrapeptide hydrogels to augment immunogenicity of the SARS-CoV-2 RBD, demonstrating improvements in humoral and potentially cellular immunity compared to no adjuvant or alum. Overall, the case that the YAWF hydrogel is acting as an immune adjuvant has been made. I was unclear how this data dovetails with the rest of the paper, since there was no prediction of adjuvant activity nor deployment of algorithmic approaches to improve this activity.]

We thank the reviewer for the comments and the suggestions to improve our work!

As the reviewer has mentioned that our manuscript details approach to improve the predictability of the behavior of self-assembling peptide hydrogels. In this work, we focus on predicting and discovering tetrapeptides for hydrogelation with an improved hit rate. Self-assembling peptides have been used as immune adjuvants due to their good biocompatibility and multivalency, so it is reasonable to believe that some tetrapeptide hydrogels we discovered could also be used to augment immunogenicity of the SARS-CoV-2 RBD, with a view to broad the biomedical applications of the natural tetrapeptide hydrogels we explored.

The reviewer considered that no algorithmic approaches has been developed to predict immune adjuvant activity in immunology experiments. To the best of our knowledge, self-assembled short peptide materials can form various nanostructures (not just nanofibers), and the adjuvant activity is tightly associated with the nanostructure and active site of the peptide, thus it may require more short peptide sequences and animal experiment data to develop the proper algorithms. Here, we chose YAWF as our model peptide to demonstrate that this type of natural tetrapeptide could be utilized as immune adjuvant. For sure, we agree with the reviewer's suggestion, and we also believe

that the uncovering of certain function between amino acid sequence and related immunogenicity would provide more convinced inspiration for further vaccine design. Therefore, our future work will focus on developing new algorithms approaches for predicting the activity of immune adjuvants based on the score function AP_{HC}. Following the suggestion of the reviewer, we revised this manuscript accordingly.

Major Comments:

[1 There is limited explanation of the immunisation model selected. Why subcutaneous dosing, why 3 doses so quickly together, why 15 µg? For comparison, human COVID vaccines were given intramuscularly 3-4 weeks apart, with immunogenicity generally assessed 2 weeks after final dose.]

We thank the reviewer for the comment. We injected the prepared vaccine subcutaneously in the groin of the mice, because the nearby lymph nodes can recognize the antigen more efficiently. The structure of tissues and organs related to the immune system between humans and mice shows functional differences, such as skin, spleen and thymus⁷. We know that memory B cells are produced after the initial immunization, and the antibodies produced after the initial immunization are mostly the unstable IgM. In the second immunization, more memory B cells were produced, while in the third immunization, the antibodies produced by B cells were mostly other relatively stable subclasses. The immunization interval of mice is generally 2-3 weeks. However, we used the monomeric RBD antigen in this work, considering that its biodegradability and epitope conformation stability were different from that of the dimeric RBD^{8,9}, so we added another injection after the first week¹⁰⁻¹².

The molecular weight of the RBD protein used in this work was 50 kDa, which was determined to be 15 µg per mouse after considering the method of inoculation, the literature, and the weight of the mice¹³⁻¹⁵.

[2 The ELISA data appear inconsistent. The alum and hydrogel groups appear to elicit comparable levels of total IgG, but alum elicited markedly less of each IgG subclass. Where is the total IgG signal for the alum animals coming from then?]

We thank the reviewer for the comment. The endpoint titres of Alum and YAWF have similar titers for IgG1, while for IgG2b and 2c, the titers of Alum group are significantly lower than YAWF (as shown in Fig. S27). Among the various IgG subclasses, IgG1 accounted for the largest proportion, and the remaining subclasses existed in small amounts. This also leads to the difference in optical density caused by different chromogenic sensitivities when we use ELISA to detect the endpoint titers of diverse IgG subclasses. We did not test the remaining IgG subclasses (such as IgG2a, IgG3, etc.), because the current data can already illustrate the general trend of YAWF promoting IgG secretion.

[3 Line 303 – “aluminum could enhance the generation of IgG by 20.7-fold. The hydrogel formed by YAWF remarkably increased the generation of IgG by 41.6-fold”. Where are these numbers coming from, how were they derived? I assume these are endpoint titres?]

We thank the reviewer for the comment. The titer of RBD-specific IgG antibodies in serum were shown in the Figure S27, the endpoint titres of RBD, aluminum and YAWF were 1407.7, 29117.5 and 58615.2, respectively. Compared with RBD group, aluminum could enhance the generation of IgG by 20.7-fold, YAWF increased the generation of IgG by 41.6-fold. Following the suggestion of the reviewer, we have added the description at Line 303 and 310 of this main text.

Revision:

Compared with RBD group, the results (Fig. 5b) showed that FDA (U.S. Food and Drug Administration) approved adjuvant aluminum could enhance the generation of IgG by 20.7-fold, and the hydrogel formed by **YAWF** remarkably increased the generation of IgG by 41.6-fold (the endpoint titres of RBD, aluminum and YAWF were shown in Fig. S27), suggesting that the tetrapeptide hydrogel could boost the immune response *in vivo*.

As for IgG2c, the hydrogel formed by **YAWF** maintained high IgG2c titers, surpassing the ones in aluminum group or control group (Fig. 5b and S27).

[4 Line 316 - "Thus, the YAWF stimulated intense T cell dependent adaptive immune response." This is not supported by the data and the language should be moderated. The authors measured cytokines in bulk splenocyte cultures. As such, the source of the cytokines cannot be automatically assigned to T cells.]

We thank the reviewer for the comment. We did not actually perform T-cell sorting on splenocytes extracted from mice, Following the suggestion of the reviewer, we corrected the description in Line 316 of this main text.

Revision:

Compared with aluminum adjuvant group, the mice received **YAWF** based vaccine showed a higher IL-5 level in their splenocytes culture, and IFN- γ secretion was also obviously evoked (Fig. 5c). Thus, the **YAWF** stimulated an obvious cell-dependent adaptive immune response. To further confirm the capability of tetrapeptide vaccine to regulate related cell immunity, the upstream dendritic cells (DCs) activation enhanced by tetrapeptide hydrogel was evaluated.

[5 Line 319 - "The DCs treated with YAWF vaccine showed promising activation as the percentage of CD83, CD80, CD86 expressing cell augmented to 72.0%, 71.1% and 50.5% (Fig. 5e)." - The DC cultures treated with RBD protein alone showed significant activation too, as well as robust TNF and IL-6 production. Why is this the case in presumably unvaccinated animals, where addition of a simple protein should be relatively inert in culture. Gating for the medium only controls should be provided for comparison.]

We thank the reviewer for the comment. Since the RBD protein itself is active against immune cells in some references^{16,17}, *in vitro* cell culture, RBD protein is directly in contact with DCs. At this time, the concentration of RBD in cell culture medium is different from that of RBD exposed to immune cells after vaccination into mice, which can effectively interact with DC. However, Free RBD is unstable *in vivo* and cannot activate the immune response effectively. Moreover, the

nanostructure formed by the tetrapeptide-based hydrogel vaccine and its slow-release effect can increase the total uptake of RBD protein by antigen-presenting cells (APCs) in lymph nodes.

Before the gating, we have analyzed the BMDCs treated with blank medium or cytokines-added medium (containing GM-CSF, IL-4) by flow cytometry, and used them as negative controls. Regardless of the cytokine addition, all the negative groups showed no significant expression of targeted CD molecules. As suggested by the reviewer, we have added the flow cytometry analysis of BMDCs in the control medium group in Fig. S28.

Revision in main text:

The DCs treated with **YAWF** vaccine showed promising activation as the percentage of CD83, CD80 and CD86 expressing cell augmented to 72.0%, 71.1% and 50.5% (Fig. 5e and S28). Such intense activation could also be proved by the clustering of DCs (Fig. 5d) producing raised levels of Th-1 cytokines (Fig. 5f).

Addition in Supplementary Information:

Fig. S28 Flow cytometry analysis of BMDCs of medium group expressing CD83, CD80, and CD86.

[6 – Groups were compared using parametric *t* tests, which are not appropriate for small animal studies ($N=6$) or tissue culture experiments ($N=3$) where normalcy of the data cannot be established.]

We thank the reviewer for pointing out this mistake. We have modified all statistical analysis method using one-way ANOVA test in Fig. 5b, 5c, 5f, and S27.

Revision:

Fig. 5 Response to tetrapeptide-based hydrogel nano vaccination. **a**) 6–8 weeks C57BL/6 mice were immunized thrice at day 0, 7, and 14 with 15 μ g RBD (RBD group), 12.5 μ L aluminum adjuvant and 15 μ g RBD (Alum + RBD group), 60 mM tetrapeptide hydrogel and 15 μ g RBD (YAWF+ RBD group). Serum and splenocytes were collected at day 21. **b**) ELISA responses to serum samples (RBD-specific) at different dilutions. SARS-CoV-2 RBD-specific IgG antibodies (IgG, IgG1, IgG2b, and IgG2c) were analyzed by endpoint dilution ELISA and measured as absorbance at 450 nm. The data were shown as the mean \pm SEM ($n=6$), and differences between RBD and other treatments were determined using one-way ANOVA test. **c**) 7 days after the last immunization, splenocytes were collected and re-stimulated with RBD protein. Bars shown were mean \pm SEM ($n=6$), and differences between RBD and other treatments are determined using one-way ANOVA test. The secretion of IL-5 and IFN- γ in the splenocytes supernatants were detected using ELISA. **d**) Optical images of bone marrow derived dendritic cells (BMDCs) treated with RBD loaded tetrapeptide hydrogel (scale bar = 100 μ m). **e**) Flow cytometry analysis of BMDCs expressing CD83, CD80 and CD86. **f**) The level of IL-6 and TNF- α in BMDCs culture supernatants were analyzed using ELISA. The data were shown as the mean \pm SEM ($n=3$), and differences between RBD and other treatments were determined using one-way ANOVA test.

Fig. S27 The titer of RBD-specific IgG₂, IgG1, IgG2b, and IgG2c antibodies in serum samples on day 21 were quantified by enzyme-linked immunosorbent assay (ELISA). The data were shown as the mean \pm SEM (n=6), and differences between RBD and other treatments were determined using one-way ANOVA test.

C) Reviewer 3

[The manuscript by Wang et al. demonstrated an integrated approach to building a score function for predicting and discovering tetrapeptide hydrogels. The approach combined coarse-grained molecular dynamics, machine learning and experiments, which built a reliable score function APHC through mutual information exchange between machine learning and experimental gelation feasibility. A remarkable gelation hit rate of 87.1% was achieved. Furthermore, as an application of the hydrogelation laws discovered, a tetrapeptide hydrogel successfully boosted the immune response of the RBD of SARS-CoV-2 in a mice model as a COVID-19 vaccine adjuvant. This work reveals may lead to further understanding self-assembly induced hydrogelation. The combine of simulation and experiments to formulate a design principle for hydrogel would greatly accelerate the biomedical applications of peptide hydrogels. Therefore, I recommend acceptance of this manuscript to Nature Communications after addressing the following issues.]

We are grateful for the comprehensive comments of the reviewer. Following the suggestion of the reviewer, we revised this manuscript accordingly.

[1. It would be more comprehensive if the authors explain why the gelation hit rate within the top 8000 AP_{HC} rank was selected as a crucial parameter when evaluating the performance of the score function.]

We thank the reviewer for the comment. The goal of this research is to develop a library within which the peptide sequences are able to form hydrogels. Therefore, we try to minimize the number of peptide candidates within the complete sequence space of tetrapeptides (# 160,000) and select 8,000 (5%) as a criterion for assessing the gelation hit rate. As a matter of fact, the exact number of hydrogel-forming peptides within the 160,000-sequence pool is unknown. However, through our “human-in-the-loop” approach, we are able to successfully classify the hydrogel-forming peptides to the 8,000 candidates pool with gelation hit rate of 87.1%. The selection of the assessing pool can be 10,000, or even 20,000, and we test the gelation hit rate within top 10,000 and top 20,000 AP_{HC} , which are 86% and 82.9% respectively. These numbers provide strong evidence that the 10,000 and 20,000 pools have “relatively more” non-hydrogel-forming peptides than that in the 8,000 pools. Since the 8,000-sequence pool is already quite large in size, we think selection of top 8,000 AP_{HC} scores for assessing the gelation hit rate is appropriate, representing a good balance between the size of hydrogel-forming peptide library and prediction accuracy. As suggested by the reviewer, we have added more discussion in Line 119-120 of this main text.

Revision:

Distinctive from all available score functions focusing on the prediction of peptide self-assembly³³, we constructed a corrected score function AP_{HC} within three loops (Fig. 2d-f) for improving the gelation hit rate. Since the final goal was to develop a hydrogel-forming peptide library with minimum candidate numbers and highest gelation possibility, we constrained our gelation hit rate assessment within the top 8,000 assessing scores (AP_H and AP_{HC}). We calculated AP_H (Fig. 1b) in the first loop and randomly selected 55 peptides (26 peptides that were among the top 8,000 in the AP_H ranking), which were possibly to form hydrogel according to human expertise.

[2. In Figure 4c, S is more common at positions 1, 2 and 4, while the authors claimed that it is beneficial for gelation when S is located at positions 1, 2 and 3. Please explain this.]

We thank the reviewer for pointing out this mistake. Following the suggestion of the reviewer, we corrected this mistake in Line 256 of this main text.

Revision:

S and **T** with moderate polarity were beneficial for gelation when **S** was located at positions 1, 2, and 4 and **T** at 1, 2.

[3. The authors are encouraged to discuss the advantages of investigating tetrapeptides for hydrogelation. How about tripeptides or pentapeptides or even polypeptides?]

We thank the reviewer for the comment. In order to systematically design short peptide hydrogels, we use natural tetrapeptides as the basic material. Compared with natural dipeptides (400 peptides) and tripeptides (80,000 peptides), 160,000 natural tetrapeptides have structural and sequence diversity to avoid traditional peptide sequence design, and their modest number of sequences also saves time for computational simulations. Although pentapeptides (320,000 peptides) and even polypeptides have more diverse structures and quantities, their computational simulations require

greater computing power for generating training data. To explore the self-assembling or hydrogelation of polypeptides or even proteins, more advanced ML or simulation technique need to be employed/developed, which falls out of the scope of this research, aiming for developing a “Human-in-the-loop” scheme for enhancing the gelation hit rate. Following the suggestion of the reviewer, we have added more discussion of tetrapeptides in Lines 63-65 of this main text.

Addition:

This work provides an integrated computation, experiment, and ML approach to build a score function for discovering tetrapeptides for hydrogelation with an improved hit rate. Tetrapeptides have sufficient structural and sequence diversity for developing a peptide hydrogel library with ample candidates, while requiring a moderate workload of simulation for generating training data. This approach proceeds as follows, firstly, the computation adopts CGMD and ML-trained regression model to provide an estimation of AP (Fig. 1a).

[4. If the C- and N- terminals are covered with functional groups or motifs, could the hydrogelation performance be possible to be predicted?]

We thank the reviewer for pointing out one possible direction of this research. Since the training data used in this work are based on non-covered peptides, the hydrogelation performance of peptides with covered C- and N-termini thus cannot be predicted using the existing models developed in this work. To perform this prediction, one has to train a new model with terminal-covered peptides, which can be a promising research direction in the future. The procedure established in this paper can be a useful guide for this work: one can first perform self-assembling simulations with different peptides covered by functional groups or motifs, train ML models, and predict the self-assembling peptides. The predicted results can then be passed to experimentalists for verification, train classification models, and predict hydrogelation performance of new terminal-covered peptide sequences. As suggested by the reviewer, we have added more discussion in Line 358 of this main text.

Addition:

The framework described here can also be extended to efficient design of other functional materials/devices, including the terminal-covered peptide hydrogels, peptide batteries, peptide fluorescence probes, and peptide semiconductors, contributing to modern organic nanotechnology employing short peptide building blocks as key structural and functional elements.

[5. I found the title could be more precise. What is the meaning of “human-in-the-loop”, experiment data or supervised learning?]

We thank the reviewer for the comment. “Human-in-the-loop” has been adopted in the field of Artificial Intelligence (AI), denoting AI systems in which human and machine jointly contribute to improving the accuracy of overall results and accelerate the learning process. Such AI systems usually involve a continuous interaction between the human and the machine in order to train a model and then monitor and update it once its deployed. In our case, we iteratively update the classification model using experimental data of gelation results, to improve the performance of

classification model in prediction of gelation or not. “Human-in-the-loop” in the title is denoting a methodology that we adopt in this research.

[6. How does AP_{HC} correlate with the properties of tetrapeptides, for example, isoelectric points?]

We thank the reviewer for the comment. To find the relation between the AP_{HC} and the isoelectric points (IP), we randomly select 20,000 tetrapeptides from the complete sequence space and calculate the IP through the online tool¹⁸: Isoelectric Point Calculator 2.0 (IPC 2.0 - Isoelectric point and pKa prediction for proteins and peptides using deep learning (mimuw.edu.pl)). The relation between the AP_{HC} and the IP is shown in Fig. S5c. There is no strong relation can be observed, however, the peptides with IP in the range of 4.5 to 6 (in pH scale) seem to have the highest AP_{HC} , indicating that peptides deprotonated (*i.e.*, negatively charged) are prone to form hydrogels. This could be explained by that, more formation of hydrogen bonds and Columbic interaction between water and peptides induce the formation of water-containing networks, if negatively charged peptides are dissolved in water solvent. In view of the factors that determine the hydrogelation of tetrapeptides, we tend to think that AP_{HC} is not only related to one factor such as hydrophilicity or isoelectric point, but a function of plenty of thermodynamic and kinetic factors. In addition, since the AP_{HC} is calculated based on AP, hydrophilicity ($\log P$), and gelation correction (C_g), we also exhibit the relation of each pair (AP_{HC} -AP and AP_{HC} - C_g) in Fig. S5a and b. The AP_{HC} - $\log P$ relation is already shown as Fig. 2i in original manuscript.

According to the reviewer's suggestion, we have added Fig. S5 to the Supplementary Information and added a supplementary description in lines 149-151 of this main text.

Addition:

Fig. S5 Correlation between a) AP_{HC} and AP, b) AP_{HC} and C_g , c) AP_{HC} and isoelectric points (IP). The IP is calculated by the online tool: Isoelectric Point Calculator 2.0 (IPC 2.0 - Isoelectric point and pKa prediction for proteins and peptides using deep learning (mimuw.edu.pl)).

Revision:

To further differentiate between AP_{HC} and AP_H in predicting peptide hydrogels, we next compared the relationship between AP_{HC} and $\log P'$ (Fig. 2i) as well as AP_H and $\log P'$ (Fig. 2j) of experimental synthesized 165 peptides that are marked with blue (gelation: yes) or red (gelation: no) dots, and those of total tetrapeptides (grey dots). Here, $\log P'$ indicated normalized hydrophilicity between 0 and 1. In addition to the AP_{HC} and $\log P'$ relation, the relation of AP_{HC} -AP and AP_{HC} - C_g were also investigated (Fig. S5a and b). No linear correlation for AP_{HC} and $\log P'$ (also AP_{HC} and AP) can be observed, demonstrating that hydrophobicity and aggregation

propensity were not the only two contributors to gelation, for instance, lower isoelectric points (*i.e.*, 4.5 ~ 6 in pH scale) could improve the gelation performance (Fig. S5c) due to the Columbic interaction and hydrogen bonds, inducing the formation of water-containing networks between deprotonated peptides and water solvent. These results indicated the significance of cooperating experimental input (*i.e.*, C_g) into prediction of hydrogel-forming sequences. Furthermore, it was conducive for hydrogelation when $\log P'$ was within the range of 0.05 to 0.4, as evidenced that the $\log P'$ of all gelating peptides were within this range (Fig. 2i).

[7. What are the pH values of the hydrogels formed? I would suggest the authors add that information in the captions of proper figures.]

We thank the reviewer for the comment. Following the suggestion of the reviewer, we have added the information of pH values of tetrapeptide hydrogels/solutions in the captions of Fig. 3a and d.

Addition:

Fig. 3 Experimental investigations on self-assembly behavior of 165 synthetic tetrapeptides.

a) TEM images of 6 representative hydrogels of synthetic tetrapeptides, respectively. Inserts: optical images of the corresponding hydrogel (pH between 7.0 to 7.5). MD simulation results (1,250 ns) and AP_{HC} ranking were shown in the right column. **b)** Dynamic frequency sweep of tetrapeptide hydrogels at the strain value of 0.5%. **c)** FTIR spectra in the amide I region of tetrapeptide hydrogels. **d)** TEM images of 6 representative non-hydrogels of tetrapeptide. Inserts: optical images of corresponding solution/suspension (pH=7.5). MD simulation results (1,250 ns) and AP_{HC} ranking were shown in the right column. **e)** Statistics and classification of morphologies obtained by TEM for hydrogel-forming tetrapeptides (100 peptides). **f)** Statistics and classification of morphologies obtained by TEM for non-hydrogel-forming tetrapeptides (65 peptides).

[8. Is it surprising that both QQQQ and EEEE form suspension at 120 mM? Any explanation from either ML perspective or thermodynamic prospective?]

We thank the reviewer for the comment. From the ML or simulation perspective, both QQQQ and EEEE cannot form suspension, suggested by that the predicted AP of QQQQ and EEEE approximately equals to 1 (*i.e.*, do not aggregate), which can be attributed to the strong hydrophilicity and lack of sufficient directional hydrogen bonds captured in simulation and prediction. However, the formation of suspension in real experiments indicate the formation of aggregates, and we attribute this to the thermodynamic factor of concentration.

Actually, we also found this phenomenon on RRRR (also contains four charged amino acids), and the MD simulation results showed that it could not aggregate, but in the actual experiments, we found the formation of aggregates of RRRR at a concentration of 120 mM (as shown in Fig. 3d). We compared the optical images of QQQQ and EEEE at four concentrations (30, 60, 90, and 120 mM, pH = 7.5) and found that QQQQ was a clear solution at 30 mM, while the solution transparency decreased (60, 90, and 120 mM) due to the formation of aggregates. As for EEEE, at 30, 60, and 90 mM, it formed uniform and clear solutions, while at 120 mM, it formed an opaque suspension. We tend to think that as the concentration of charged tetrapeptides increases, the hydrogen bond between peptide molecules and water is strengthened, resulting in the formation of

aggregates at high concentrations. According to the reviewer's suggestion, we have modified the description in Lines 214-216 of this main text.

Fig. R1 The optical images of QQQQ and EEEE aqueous solutions at four concentrations (30, 60, 90, and 120 mM, pH = 7.5).

Revision:

TEM images (Fig. 3d and Extended Data Fig. 5) showed that these six peptides formed aggregates with different sizes in an aqueous solution, qualitatively consistent with the morphologies obtained in MD simulations except for **RRRR** (Fig. 3d, right column), showing different levels of aggregation. Taking the TEM result of **RRRR** together, we attributed this to the thermodynamic factor of concentration. Finally, we presented a summary of the assembled morphologies of all synthesized tetrapeptides (Fig. 3e, 3f and Extended Data Fig. 5), indicating that hydrogel-forming tetrapeptides tended to form fibers, sheets, or hybrid morphology (70%) in an aqueous solution.

Reference:

- 1 Mitchell, W. *et al.* Structure-activity relationships of mitochondria-targeted tetrapeptide pharmacological compounds. *Elife* **11** (2022).
- 2 Yan, J. K. *et al.* Acute myocardial infarction therapy using calycosin and tanshinone co-loaded; mitochondrion-targeted tetrapeptide and cyclic arginyl-glycyl-aspartic acid peptide co-modified lipid-polymer hybrid nano-system: preparation, characterization, and anti myocardial infarction activity assessment. *Drug Delivery* **29**, 2815-2823 (2022).
- 3 Ericson, M. D. *et al.* Functional Mixture-Based Positional Scan Identifies a Library of Antagonist Tetrapeptide Sequences (LAtTeS) with Nanomolar Potency for the Melanocortin-4 Receptor and Equipotent with the Endogenous AGRP(86-132) Antagonist. *J. Med. Chem.* **64**, 14860-14875 (2021).

- 4 Zhang, J. R. *et al.* Neuroprotective effects of maize tetrapeptide-anchored gold nanoparticles in Alzheimer's disease. *COLLOIDS AND SURFACES B-BIOINTERFACES* **200** (2021).
- 5 Lee, S., Peng, J., Williams, A. & Shin, D. ASCENDS: advanced data SCiENce toolkit for non-data scientists. *Journal of Open Source Software* **5** (2020).
- 6 Hsu, C.-W. *et al.* A practical guide to support vector classification. 1396-1400 (2003).
- 7 Medetgul-Ernar, K. & Davis, M. M. Standing on the shoulders of mice. *Immunity* **55**, 1343-1353 (2022).
- 8 Dai, L. *et al.* Efficacy and Safety of the RBD-Dimer-Based Covid-19 Vaccine ZF2001 in Adults. *N. Engl. J. Med.* (2022).
- 9 Sun, S. *et al.* Recombinant vaccine containing an RBD-Fc fusion induced protection against SARS-CoV-2 in nonhuman primates and mice. *Cell. Mol. Immunol.* **18**, 1070-1073 (2021).
- 10 Bechinger, B. *J. Membr. Biol.* **156**, 211 (1997).
- 11 Verma, V. *et al.* PD-1 blockade in subprimed CD8 cells induces dysfunctional PD-1(+)CD38(hi) cells and anti-PD-1 resistance. *Nat. Immunol.* **20**, 1231-+ (2019).
- 12 Wang, S. H., Gao, J., Li, M., Wang, L. G. & Wang, Z. J. A facile approach for development of a vaccine made of bacterial double-layered membrane vesicles (DMVs). *Biomaterials* **187**, 28-38 (2018).
- 13 Wang, Z. Z. *et al.* Exosomes decorated with a recombinant SARS-CoV-2 receptor-binding domain as an inhalable COVID-19 vaccine. *Nat. Biomed. Eng.* **6** (2022).
- 14 Zhang, Y. *et al.* Amantadine-assembled nanostimulator enhances dimeric RBD antigen-elicited cross-neutralization against SARS-CoV-2 strains. *Nano Today* **43**, 101393 (2022).
- 15 Cong, M. Q. *et al.* 3-dose of RBD vaccine is sufficient to elicit a long-lasting memory response against SARS-CoV-2 infection. *Signal Transduct Tar* **7** (2022).
- 16 Zhou, R. *et al.* Acute SARS-CoV-2 Infection Impairs Dendritic Cell and T Cell Responses. *Immunity* **53**, 864-877.e865 (2020).
- 17 Zhao, Y. *et al.* SARS-CoV-2 spike protein interacts with and activates TLR41. *Cell Res.* **31**, 818-820 (2021).
- 18 Kozlowski, Lukasz P. IPC 2.0: prediction of isoelectric point and pKa dissociation constants. *Nucleic Acids Res.* **49**, W285-W292 (2021).

REVIEWER COMMENTS

Reviewer #1 (Remarks to the Author):

The authors have done a great job to address my concerns. I recommend this paper for acceptance.

Reviewer #3 (Remarks to the Author):

The authors have addressed my previous concerns, and I support the acceptance of this topical work.

Reviewer #4 (Remarks to the Author):

In this paper, the authors study how tetrapeptides can form hydrogels. The authors combine MD simulations, machine learning and experimental testing in a novel way. Each component might not be novel, but the combination improves the paper.

I did not review the first version of the paper, so I base my review on the revised version and the comments to the reviewers of the original version.

The machine learning part is relatively standardized and might benefit from shortening. The main manuscript does not need to include detailed descriptions of one-hot encoding and other coding schemes. I do not see much-added value in comparing the different machine learning methods (it is clear that the SVMs are best). However, as pointed out by reviewer #1, the comparison with simpler encoding (dipeptide or amino acid frequencies) is of interest. Here, I think the authors have misunderstood the meaning of these comparisons. They are not primarily aimed at examining the best possible way to predict the properties of the peptides, but to understand position-specific rules. Unfortunately, using the 1200-bit version does not enable such a comparison (as this is just a cumbersome way to encode the 80-bit one-hot encoding. Instead, a 400 bit dipeptide and a 20-bit AAfrequency encoding should be tried and discussed. However, I think there might be easier ways to extract the relevant information by simple statistics on the dipeptides. Some simple log-odds plots on AAfrequency (and secondary features such as pi-stacking potential) of the experimentally verified (or predicted) hydrogel forming peptides should provide the sought after information.

The following are our responses to the comments (in italics) of the reviewer 4 and the changes (underlined) in the manuscripts.

Reviewer 4

[In this paper, the authors study how tetrapeptides can form hydrogels. The authors combine MD simulations, machine learning and experimental testing in a novel way. Each component might not be novel, but the combination improves the paper.]

I did not review the first version of the paper, so I base my review on the revised version and the comments to the reviewers of the original version.]

We thank the reviewer for the positive comments, and we have addressed all the points raised by the reviewer 4 accordingly.

[1. The machine learning part is relatively standardized and might benefit from shortening. The main manuscript does not need to include detailed descriptions of one-hot encoding and other coding schemes.]

We thank the reviewer for the comments to improve the clarity of the manuscript. For the machine learning part, the details of the training and testing under different conditions, including the training algorithms, hyperparameters, feature representation approaches, and the size of the training datasets, have been relegated to the supplementary file. In the main manuscript, we only present the necessary information on model training and performance in a single sentence [*i.e.*, Using the algorithm of support vector machine (SVM)³⁸ with 10,000 training data represented by 80-bit one-hot approach with amino acid sequence (Tab. S3), we obtained a reliable SVM model with training/testing performance of 0.095/0.092 in mean absolute difference (MAE_{tr}/MAE_{te}) and 0.928/0.933 in coefficient of determination (R_{2tr}/R_{2te})³⁹ (Fig. 2a and b)], for the interested readers to reproduce the results. We have also revised the Methods section of machine learning to improve the conciseness while maintaining the necessary training, validation, and testing information. We have also shortened the description of one-hot and other coding schemes in the Methods section.

Revisions in the Methods section of “Machine learning”:

Four different ML algorithms were deployed: Random Forest (RF)⁵⁸, Linear Regression (LR)⁵⁹, Nearest Neighbor (NN)⁶⁰, and Support Vector Machine (SVM)⁶¹. Mean absolute error (MAE) and coefficient of determination (R2)⁶² were calculated to assess the performance of each ML model. Different numbers of training data sets (*i.e.*, 1,000, 5,000, and 10,000) were used to train the ML models. In each training, 80% of the training data was used for training, while the remaining 20% was used for validation (Fig. 1b). After obtaining each model, another 5,000 data were employed for independent testing.

Before training of the ML model, we converted the amino acid sequence into numerical data with 4-integer and 80 bit one-hot representation approaches (shown in Tab S3, taking Glu-His-Asn-Thr, *i.e.*, EHNT, as an example), aiming for enhanced model performance with optimal data presentation approach. Moreover, a tetrapeptide can be considered as a “tripeptide” with each “position” represented by one of the 400 possible dipeptide sequences, namely, a tetrapeptide can have 1200 possible bits with 3 of them to be 1. Therefore, we also trained models with a 1200-bit one-hot representation converted from the dipeptide sequence composition.

[2. I do not see much-added value in comparing the different machine learning methods (it is clear that the SVMs are best).]

We thank the reviewer for the comments. Based on the authors' experience with machine learning, other algorithms such as linear regression (LR) can perform even better than the SVM, since the SVM can sometimes fail and cannot be adopted for prediction, due to different training conditions [1] Wang, Jiaqi, et al. "Machine learning for thermal transport analysis of aluminum alloys with precipitate morphology." *Advanced Theory and Simulations* 2.4 (2019): 1800196; 2) Wang, Jiaqi, et al. "Interatomic Potential Model Development: Finite-Temperature Dynamics Machine Learning." *Advanced Theory and Simulations* 3.2 (2020): 1900210]. Therefore, we tested four different algorithms in this research. It can be observed that, in Fig. S1 with 1,000 training datasets, the LR can achieve a comparable performance to SVM, and as the number of training datasets increases, the training and testing performance of random forest also increases, close to that of SVM (Fig. S3). Therefore, we believe that different algorithms should be tested under various conditions to find the optimal one.

3. However, as pointed out by reviewer #1, the comparison with simpler encoding (dipeptide or amino acid frequencies) is of interest. Here, I think the authors have misunderstood the meaning of these comparisons. They are not primarily aimed at examining the best possible way to predict the properties of the peptides, but to understand position-specific rules. Unfortunately, using the 1200-bit version does not enable such a comparison (as this is just a cumbersome way to encode the 80-bit one-hot encoding). Instead, a 400 bit dipeptide and a 20-bit AAfrequency encoding should be tried and discussed. However, I think there might be easier ways to extract the relevant information by simple statistics on the dipeptides. Some simple log-odds plots on AAfrequency (and secondary features such as pi-stacking potential) of the experimentally verified (or predicted) hydrogel forming peptides should provide the sought after information.

We are grateful for the reviewer's comments for improving the understanding in position-specific rules of peptide hydrogelation. As pointed out by the reviewer 1: "*if would be better if the authors developed the model using amino acid and dipeptide composition encoding, see whether they improve the prediction accuracy*", we then employed 4-integer encoding, 80-bit one-hot encoding with amino acid composition, and 1200-bit one-hot encoding with dipeptide composition to compare the prediction accuracy of the trained models. However, as pointed by reviewer 4, we did not try to generate the position-specific rules by different encoding approaches.

In terms of position-specific rules, **we have generated the position-specific hydrogelation laws in Fig.4c in the original manuscript, by comparing the percentage of the amino acids in each position** (*i.e.*, this can be taken as a statistical approach), for both experimentally verified and computationally predicted hydrogel-forming peptides, and a satisfactory agreement has been reached. To extend the hydrogelation laws with respect to dipeptide and tripeptide compositions, we plotted 20 figures (Fig. 4d and Fig. S7-S25) and in each figure, one amino acid is fixed at C-terminus, while the *x*-axis indicates an amino acid in position 1 (N-terminus), *y*-axis indicates an amino acid in position 2, and third position is as shown in the rectangular box (Fig. 4d). These hydrogelation laws have been detailed in the section of "Hydrogelation laws from experiment and simulation results".

Following the reviewers' comments, we also tried the encoding of the tetrapeptides with 400-bit dipeptide sequence with 100 experimental data and 8,000 top AP_{HC} values, and explored the position-specific rules, as a supplementary to the hydrogelation rules with respect to dipeptide composition, as shown in Figure S26.

Revisions in the Supplementary Information:

Figure S26: Percentage of amino acid pairs (*i.e.*, dipeptide) of 100 hydrogel-forming peptides and 8,000 peptides with top AP_{HC} score in simulations.

We added the following sentences in the section “Hydrogelation laws from experiment and simulation results” of the revised manuscript:

Revisions in the main text:

In addition, aromatic amino acids bonded with **P** and **K** exhibited similar positive performance. These rules can also be applied to the triplets. In addition, we analyzed the position-type percentage with respect to adjacent amino acids, based on the 100 hydrogel-forming peptides in the experiment and 8,000 peptides with the highest AP_{HC} score in the simulation (Fig. S26). It can also be deduced that aromatic-aromatic and aromatic-hydrophobic doublets have the most significant contribution to hydrogelation, and position-specific rules regarding other amino acids are also congruent with those deduced from Fig. 4c and d. For example, amino acid **A** is barely found in the fourth position, except that when **F** or **Y** is located in the third position. In summary, we have presented a complete picture of the relationship between the gelation ability and position & type of 20 natural amino acids, providing a schematic guidance for experimentalists to design tetrapeptide hydrogels and possibly functional applications associated.

REVIEWERS' COMMENTS

Reviewer #4 (Remarks to the Author):

The authors have satisfactory answered all my concerned